# Computational investigation of *cis*-1,4-polyisoprene binding to the latex-clearing protein LcpK30

**Aziana Abu Hassan, Marko Hanževački**[¤]*, **Anca Pordea**[*]

Faculty of Engineering, University of Nottingham, Nottingham, United Kingdom

¤ Current address: School of Chemistry, University of Bristol, Bristol, United Kingdom
* anca.pordea@nottingham.ac.uk (AP); marko.hanzevacki@bristol.ac.uk (MH)

**Data Availability Statement:** Input and output information from the computational analysis including Caver and fpocket setups; GOLD docking proto-cols; additional analysis from MD simulations and full MD inputs; and details on

## Abstract

Latex clearing proteins (Lcps) catalyze the oxidative cleavage of the C = C bonds in *cis*-1,4-polyisoprene (natural rubber), producing oligomeric compounds that can be repurposed to other materials. The active catalytic site of Lcps is buried inside the protein structure, thus raising the question of how the large hydrophobic rubber chains can access the catalytic center. To improve our understanding of hydrophobic polymeric substrate binding to Lcps and subsequent catalysis, we investigated the interaction of a substrate model containing ten carbon-carbon double bonds with the structurally characterized LcpK30, using multiple computational tools. Prediction of the putative tunnels and cavities in the LcpK30 structure, using CAVER-Pymol plugin 3.0.3, fpocket and Molecular Dynamic (MD) simulations provided valuable insights on how substrate enters from the surface to the buried active site. Two dominant tunnels were discovered that provided feasible routes for substrate binding, and the presence of two hydrophobic pockets was predicted near the heme cofactor. The larger of these pockets is likely to accommodate the substrate and to determine the size distribution of the oligomers. Protein-ligand docking was carried out using GOLD software to predict the conformations and interactions of the substrate within the protein active site. Deeper insight into the protein-substrate interactions, including close-contacts, binding energies and potential cleavage sites in the *cis*-1,4-polyisoprene, were obtained from MD simulations. Our findings provide further justification that the protein-substrate complexation in LcpK30 is mainly driven by the hydrophobic interactions accompanied by mutual conformational changes of both molecules. Two potential binding modes were identified, with the substrate in either extended or folded conformations. Whilst binding in the extended conformation was most favorable, the folded conformation suggested a preference for cleavage of a central double bond, leading to a preference for oligomers with 5 to 6 C = C bonds. The results provide insight into further enzyme engineering studies to improve catalytic activity and diversify the substrate and product scope of Lcps.

protein–ligand docking results are freely available on Figshare: https://doi.org/10.6084/m9.figshare.22941149.

**Funding:** The research was funded by the Malaysian Rubber Board, who provided a studentship for AAH and financial support for the project. We also gratefully acknowledge the support and access to the University of Nottingham High Performance Computing Facility. The funders had no role in study design, data collection and analysis, decision to publish, or preparation of the manuscript.

**Competing interests:** The authors have declared that no competing interests exist.

# Introduction

Rubber materials are essential in everyday life, with more than 25 million metric tons of natural and synthetic rubber being produced and consumed yearly [1]. This results in a high amount of rubber waste, which is challenging to recycle or decompose, thus creating a serious environmental concern once the material reaches the end of life. An economically and environmentally sustainable strategy for rubber waste disposal is yet to be established. The degradation of rubber polymers mainly involves oxidative cleavage of C = C bonds in the polymer backbone and has been reported with organic or organometallic catalysts, which are limited by the use of rare metals (e.g. Grubbs catalyst) and harsh reagents [2]. An attractive alternative is enzymatic rubber degradation using oxidative enzymes. Rubber oxygenases (RoxA and RoxB), isolated from Gram-negative bacteria and latex clearing proteins (Lcps), isolated mostly from Gram-positive bacteria, have been shown to degrade both natural and synthetic *cis*-1,4-polyisoprene, with the latter enzymes being more extensively studied [3]. The products are functionalized oligoisoprenoids carrying a carbonyl group (ketone and aldehyde) at each end (Fig 1a). This cleavage preserves the polyisoprene structure and gives access to lower molecular weight products, which can be either further metabolized or used as materials with potentially interesting properties [4]. Thus, rubber degrading enzymes offer exciting perspectives for polymer degradation within a circular economy and have been extensively studied in the past few years.

A range of Lcps have been identified from several organisms [5–9] and were shown to be *b*-type cytochromes with a non-covalently bound heme cofactor. The crystal structure of LcpK30 has been determined to possess a globin-like fold, showing the heme to be coordinated

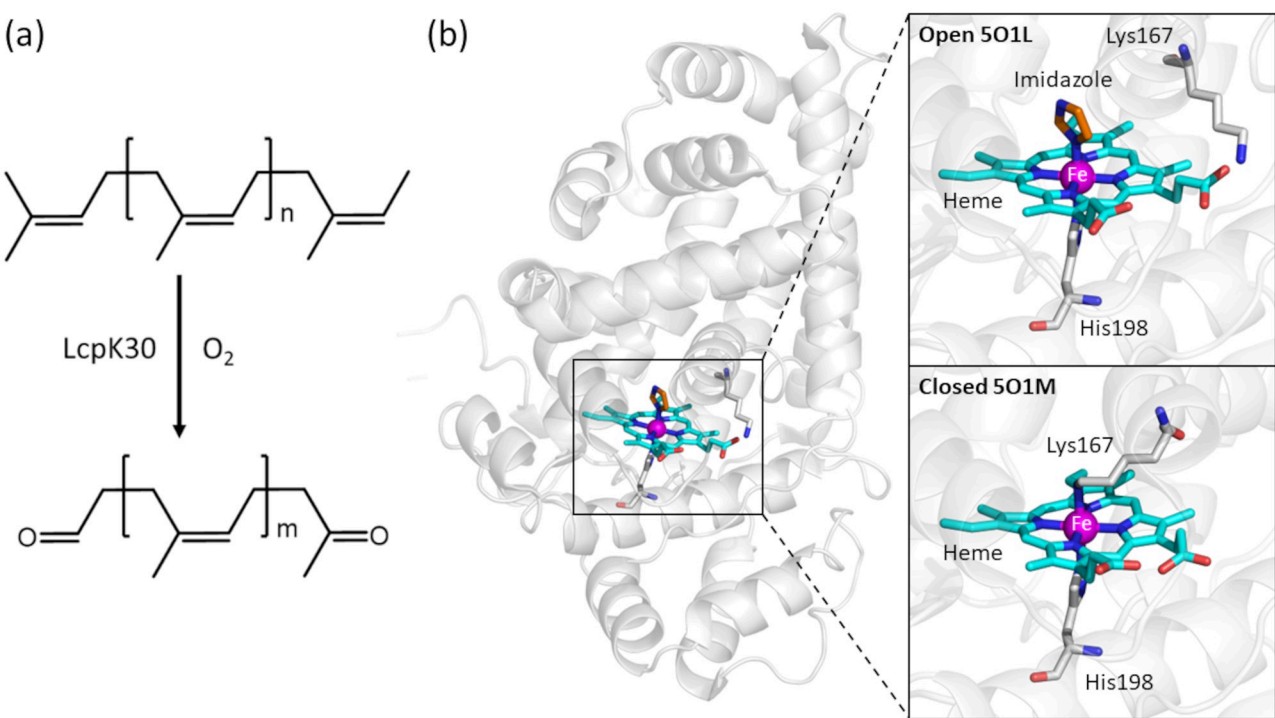

**Fig 1. Crystal structure and *cis*-1,4-polyisoprene cleavage reaction catalyzed by LcpK30.** (a) Oxidative cleavage of *cis*-1,4-polyisoprene catalyzed by LcpK30. (b) Crystal structure of LcpK30 (PDB ID: 5O1L) with a close view of heme in the active site..Coordination of an imidazole molecule to the distal site of heme fixes the protein structure in its open conformation (PDB: 5O1L), while the coordination by the sidechain of Lys167 forces the closed state of the enzyme (PDB: 5O1M).

to His198 (Fig 1b) [10]. The oxidative cleavage of polyisoprene by Lcp occurs in an endo-type mechanism and results in a mixture of oligoisoprenoids with different lengths, from C20 to C65 (3 to 12 isoprene units) [3, 11–13]. A defined distribution of degradation products is observed, with a preference for 5–6 (for Lcp1VH2) [14] or 4–7 (for LcpK30) [1, 8, 15] isoprene repetitive units and lower concentrations of smaller and larger products. Quantum mechanics/molecular mechanics (QM/MM) calculations suggested that the addition of the heme-bound dioxygen to the C = C bond triggers its cleavage via a dioxetane intermediate [16]. This mechanism is similar to two other heme-based dioxygenases, indoleamine and tryptophan 2,3-dioxygenases.

Structural and spectroscopic analysis of LcpK30 suggested the existence of two conformational states, closed and open states (Fig 1b) [10]. In the open state, a continuous hydrophobic substrate channel passes next to the heme group and allows the substrate to access the active site, whilst, in the closed state, the channel is blocked by coordination of the heme to Lys167. This difference is most visible in the conformation of the active site helix (also known as helix E) [10], comprised of residues 160–176. The helix is more structured in the closed state compared to the open state, in which this helix is split into two fragments (S1 Fig). Most enzymes that react with polymeric substrates, such as PETases (with PET substrates) and lytic polysaccharide monooxygenases, LPMOs (with polysaccharide substrates) contain exposed active sites, which allow easy contact with the polymer chains of the substrate [17, 18]. Intriguingly, the catalytic heme cofactor in Lcps is buried within a hydrophobic pocket, which raises the question of how the insoluble hydrophobic rubber chains can access the active site.

Despite an increased understanding of the structure and catalytic mechanism of Lcps, a useful process for enzymatic rubber degradation has not been achieved. The enzymatic degradation rates remain low, and the range of oligoisoprenoid products is too heterogeneous to be convenient for applications as materials. Furthermore, the enzymatic degradation is limited to *cis*-1,4-polyisoprene rubber, whilst the degradation of other diene rubbers remains largely underexplored. These challenges could potentially be addressed through protein engineering approaches, which require an understanding of the interaction between substrate and protein.

In this study, we used a computational modelling approach starting from the LcpK30 crystal structure in open conformation to understand the interaction between LcpK30 and its flexible polyisoprene substrate. Firstly, the potential access pathways of *cis*-1,4-polyisoprene to the active site of LcpK30 were explored in the static structure using the CAVER PyMol plugin 3.0.3 and fpocket, to identify relevant hydrophobic tunnels and pockets that are large enough to accommodate the substrate [19]. Dynamic simulations further investigated tunnel availability and flexibility. We then carried out supramolecular docking of a C50 substrate model containing 10 repeating isoprene units ($C_{50}H_{82}$), which was deemed sufficiently large to provide valuable insights into the location, conformations and interaction of the polymer chains with the enzyme. Given the challenge to obtain precise information from the docking of long alk (en)yl chains without specific pharmacophores, we used molecular dynamics (MD) simulations to provide a better understanding of the stability and the nature of the predicted substrate poses and conformational changes of the protein.

## Results and discussion

### Tunnel availability and conformational flexibility of LcpK30 without substrate bound

Putative tunnels were initially identified within the static crystal structure of LcpK30, to obtain a general overview of potentially important pathways for the substrate access from the surface to the active site. This was done with the CAVER-PyMOL plugin, which provides a graphical

interface for setting up the calculation and allows interactive visualization of tunnels or channels in protein static structures [20]. We used both the open (5O1L) and the closed state conformations (5O1M), however only the former led to tunnel identification using the default parameters (see Methods). A total of four tunnels were identified, and their lengths, radii, and bottlenecks (narrowest parts of the tunnels) were determined (see S1 Table and S2 Fig for details on the static tunnel features). Tunnels 1 and 2 had the highest throughput and, thus, the highest probability to be used as routes for the transport of substrates (Fig 2a and S1 Table). Near the heme cofactor in the active site, these tunnels were lined up by residues known to be of high importance for LcpK30 activity: Glu148, Arg164, Lys167 and Thr168 [10, 12]. Other, more surface-exposed residues interacting with the tunnels included Trp82, Thr83, Arg84, Ala151, Val152, Gly157, Gly158, Ala159, Asp163, Ile165, Ala166, Ala169, Arg170, Leu171, Asp174, His184, Gly185, Ser186, Val189, Thr190, Lys193, Thr194, Val197, His198, Thr230, Glu392, Gly393, Arg394, Arg395, Ile396, Ala397, Ile398, Asp399, and Pro401. The radius of putative tunnels ranged between approximately 1 Å and 2.8 Å, with the bottleneck (0.96 Å) located near the heme and the wider radius found towards the outer surface of the protein.

The average radii throughout both tunnels were smaller than the estimated width of the *cis*-1,4-polyisoprene substrate molecule (S2c Fig), which suggested that significant conformational

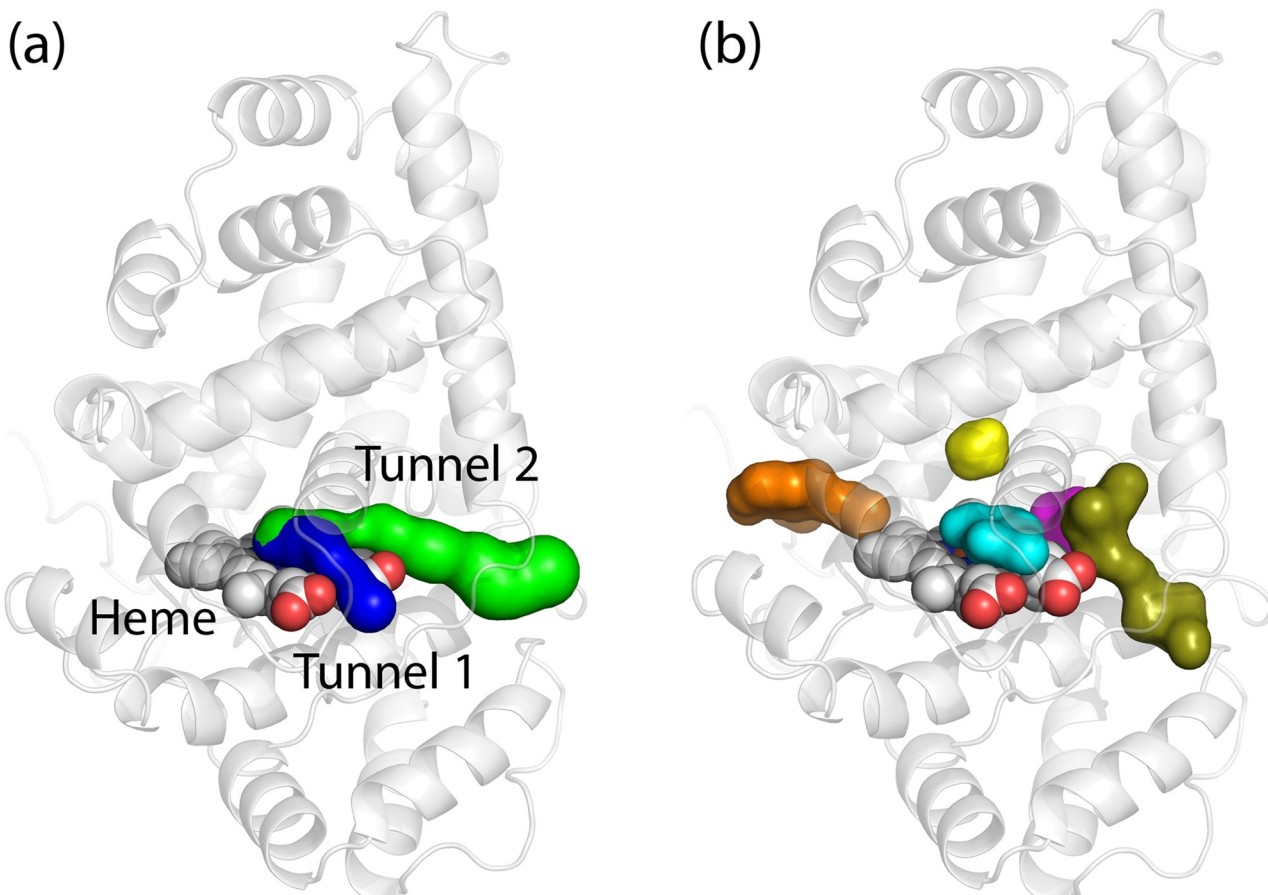

**Fig 2. Static tunnel mapping and pocket identification in the LcpK30 open structure without substrate.** (a) Two dominant tunnels identified by CAVER-Pymol plugin 3.0.3 (tunnel 1 in blue and tunnel 2 in green); (b) five detected pockets or cavities closest to heme in the LcpK30 static X-ray structure, identified by automated geometry-based fpocket tool. The pockets are pocket 1 (olive), pocket 3 (cyan), pocket 4 (orange), pocket 9 (magenta) and pocket 12 (yellow).

changes of the protein are required to accommodate the hydrophobic chains of the large polymeric substrates through this gateway. Alternatively, these tunnels could also be suited for transporting smaller oxygen to the heme. We observed a significant difference in the length of the tunnels from the heme starting point to the protein surface, with tunnel 2 being about twice as long compared to tunnel 1. Whilst a shorter tunnel such as tunnel 1 may generally give quicker access of substrates to the active site, in this case, where the substrate is a long polymer, the longer tunnel 2 may provide more opportunity for protein-substrate interaction. Furthermore, tunnel 2 has a wider opening (2.8 Å) at the protein surface (Glu392, Gly393, Arg394, Arg395, Ile396) which agrees well with the estimated thickness of *cis*-1,4-polyisoprene and which could represent a potential gateway for the substrate access to the active site.

The important pockets (protein cavities) in the enzyme in the open state were also predicted using an automated geometry-based fpocket tool that decomposes a 3D protein into Voronoi polyhedrals (see Methods for details). A total of 24 pockets were detected throughout LcpK30. However, only five pockets (pocket 1, 3, 4, 9 and 12) were considered relevant for substrate binding due to their near proximity to heme and the active site (Fig 2b). Out of these five, two pockets were especially interesting since they are located at a distal site directly above the heme cofactor: a larger pocket 3, with a pocket volume of 173.52 $Å^3$ and a smaller pocket 12, with a volume of 77.21 $Å^3$. These two pockets also had a relatively high hydrophobicity score, indicating that they could accommodate the lipophilic *cis*-1,4-polyisoprene ligand in its bound state. Interestingly, these cavities were also occupied by imidazole molecules in the crystal structure. The residues surrounding these two pockets are Ser138, Ser142, Glu148, Val152, Arg164, Lys167, Thr168, Leu171, Thr230, Ser233 and Leu234, and include the same residues lining up the tunnels identified earlier using CAVER-PyMOL plugin. Taken together, the information about the tunnels and pockets calculated on the X-ray structure of LcpK30, provided a better picture on possible entry pathways that lead from the surface into the hydrophobic active site of the protein interior and gave an indication of the space where ligand might bind.

Since the intrinsically dynamic nature of the protein could largely influence the pathways and cavities through the protein, we carried out extensive MD simulations of LcpK30 to obtain more information, specifically on the conformational changes that could influence binding of the substrate to LcpK30. We assumed that polyisoprene binding occurs in the catalytically active state of LcpK30 and therefore we prepared an LcpK30 model starting from the open state structure, in which the co-crystalized imidazole molecule was replaced with $O_2$ bound to the heme cofactor and where the Lys167 chain (not heme bound) was fully protonated, which we were able to simulate during relatively long simulation timescales. For comparison, we also modelled the closed-like state system, prepared from the closed state structure conformation, but where the coordination of Lys167 to heme was removed and replaced with $O_2$ coordination, whilst Lys167 was fully protonated.

Relatively low C-alpha backbone RMSD were obtained over the course of 500 ns MD simulations, for both open and closed-like conformations (see S3a Fig for the open conformation). This suggested that the enzyme with no substrate bound did not undergo large structural changes compared to the reference X-ray structures of the open and closed states. The RMS fluctuation analysis of the C-alpha atoms revealed conformational changes of the open state protein linked to residues 155–160 from the loop near heme and residues 392–396 at the C-terminus (S3b Fig). Similar fluctuations were also found in the enzyme from the experimental crystallographic B-factor (S4 Fig). We further characterized these conformational changes as the loop rotation containing Arg394, which occasionally replaced Lys167, forming the salt bridge with the carboxylate group of heme during the simulation. Taking into account these dynamic features of the protein, we hypothesize that the structural adjustment of LcpK30

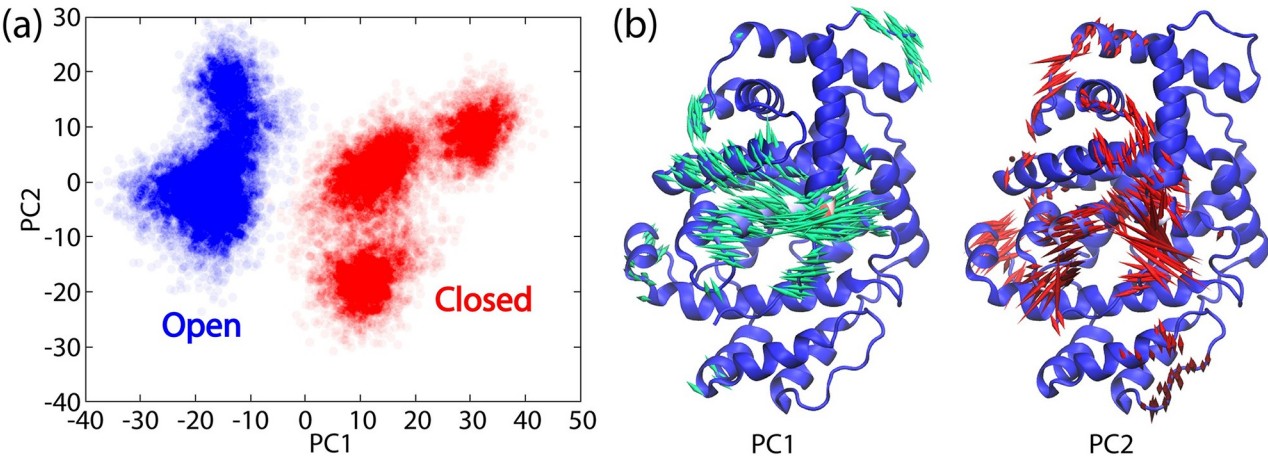

**Fig 3. Principal component analysis (PCA) of MD simulations of open and closed-like states of LcpK30 without substrate.** PCA was performed considering cartesian coordinates of protein backbone atoms (N, Cα, C and O). (a) MD snapshots projected on the first two principal components. Blue and red circles represent MD snapshots belonging to open and closed-like states, respectively. (b) Normal mode displacement vectors associated with the first two principal components showing only motions longer than 2 Å as porcupine in both directions. The backbone is shown as cartoon representation and colored by the mobility where lower and higher flexibility is depicted in blue and red, respectively.

could facilitate the entry of substrate and exit of oligoisoprenoid products through the gateway comprised of C-terminus loop 392–396 and flexible loop 155–160.

Principal component analysis (PCA) taking into account backbone cartesian coordinates was used to compare open and closed states of the enzyme with no substrate. A clear separation of the two states along the PC1 was observed, suggesting that major conformational changes must occur that would allow the transition between closed (positive PC1 values) and open (negative PC1 values) conformations (Fig 3a). Furthermore, the normal mode displacements (Fig 3b) along the PC1 were characterised as conformational changes that predominantly involve helix E residues 160–176, which also contains the central Lys167. Namely, in the open conformation Lys167 is exposed towards the protein surface interacting with the propionates of the heme cofactor, in contrast to the closed-like conformation where it is rotated into the protein interior interacting with $O_2$ and Glu148. On the other hand, displacements along the PC2 are associated with the flexibility of the C-terminus loop 392–397 ranging from fully closed (negative PC2 values) to a fully open (positive PC2 values) conformation (Fig 3b).

Despite the closed-like state having identical amino acid protonation states and the same species present in the active site as the open state (fully protonated Lys167 and $O_2$ bound to heme, a transition between the open and the closed-like states was not observed during the simulation, possibly because of a large barrier separating these two states, which simply could not be sampled using conventional MD simulations at the available timescales. This implies that the opening of the enzyme is a complex and controlled process that happens on longer timescales and involves large conformational changes of the helix E and Lys167 and might even be coupled with the approach and the interaction with the polyisoprene substrate.

Given the enzyme flexibility required to accommodate the substrate, we sought to improve the static tunnel predictions by employing the dynamical approach implemented in CAVER Analyst 2, using MD snapshots instead of static crystal structures. The first 5 identified tunnel clusters ranked based on priority (calculated by averaging tunnel throughputs over all snapshots) were analyzed, for MD simulations with both the open and closed-like conformations and the details about the features of the dynamic tunnel clusters are shown in Table 1. Open

**Table 1. Features of first 5 dominant dynamical tunnel clusters in LcpK30 open and closed-like states.**

| ID | Average bottleneck radius (Å) | Std. dev. | Maximum bottleneck radius (Å) | Average length (Å) | Std. dev. | Average curvature | Std. dev. | Average throughput | Std. dev. |
|----|----|----|----|----|----|----|----|----|----|
| | | | | Open state | | | | | |
| 1 | 1.36 | 0.21 | 1.98 | 13.79 | 3.01 | 1.34 | 0.16 | 0.64 | 0.09 |
| 2 | 1.13 | 0.18 | 1.87 | 10.68 | 2.71 | 1.18 | 0.10 | 0.60 | 0.07 |
| 3 | 1.33 | 0.21 | 1.87 | 19.43 | 3.51 | 1.28 | 0.10 | 0.56 | 0.09 |
| 4 | 1.18 | 0.17 | 1.62 | 20.43 | 3.29 | 1.38 | 0.17 | 0.46 | 0.06 |
| 5 | 1.37 | 0.21 | 1.77 | 10.72 | 2.44 | 1.20 | 0.14 | 0.66 | 0.08 |
| | | | | Closed-like state | | | | | |
| 1 | 1.01 | 0.08 | 1.21 | 14.92 | 4.13 | 1.31 | 0.20 | 0.48 | 0.11 |
| 2 | 1.00 | 0.08 | 1.19 | 19.06 | 4.65 | 1.60 | 0.34 | 0.39 | 0.08 |
| 3 | 0.99 | 0.07 | 1.20 | 17.40 | 4.72 | 1.44 | 0.33 | 0.42 | 0.09 |
| 4 | 0.98 | 0.07 | 1.16 | 23.52 | 5.09 | 1.74 | 0.31 | 0.35 | 0.09 |
| 5 | 0.98 | 0.07 | 1.20 | 24.88 | 4.84 | 1.82 | 0.33 | 0.30 | 0.08 |

Calculations were performed on 1500 snapshots from MD simulations of open (upper) and closed-like (lower) states of LcpK30.

state tunnel clusters had overall larger average and maximum bottleneck radius, shorter length, smaller curvature and higher throughput than closed state ones, which ultimately makes them more favorable for substrate binding. Furthermore, the priority of the tunnel clusters calculated based on the appearance frequency during the MD simulations was much higher in the open (14–49%) compared to the closed (4–13%) state. Interestingly, when comparing the spatial distribution of timeless tunnel cluster pathways, we could clearly see how the central Lys167 plays an important role in separating tunnels into two wider superclusters in the open state (Fig 4a, clusters 1 and 3 forming the green supercluster and clusters 2 and 5 forming the blue supercluster). This is in contrast with the rotated Lys167 conformation forcing the appearance of one narrow supercluster in the closed-like state (S5 Fig, clusters 1,3,4 and 5 forming the limon supercluster).

The two dominant favorable tunnel superclusters obtained with the open state were similar to tunnels 1 and 2 initially calculated in the static X-ray structure (Fig 1), but with a slightly larger average and maximum bottleneck radii. Both superclusters were lined by predominantly hydrophobic residues as found in the case of static tunnels. This suggests that structural changes of LcpK30 do not largely influence the ranking and features of the putative pathways for substrate access. Interestingly, we also observed a new branched pathway connecting the heme cofactor in the protein interior with the surface through the tunnels surrounded by Glu148, Ile145, Asp226, Val229 and Thr230 (cluster 4 in Fig 4a). Due to the salt bridge between Arg147 and the neighboring Glu52 and Asp226, cluster 4 splits from its main route into two pathways. Because of its smaller average bottleneck radius, this cluster might be important for the transport of smaller products after the cleavage or $O_2$ into and from the active site. We also noted that such a narrow tunnel could only be observed in the static structure after decreasing a minimum probe radius.

The analysis of potential protein cavities using the snapshots from MD simulations also confirmed the persistence of two hydrophobic pockets that could accommodate ligands in the active site near heme, both with slightly increased average volumes compared to the static structure. The larger pocket 3 directly interacts with heme and has an average volume of 182.08 $Å^3$. A smaller pocket 12 has an average volume of 116.97 $Å^3$ and is buried deeper in the protein interior (Fig 4b).

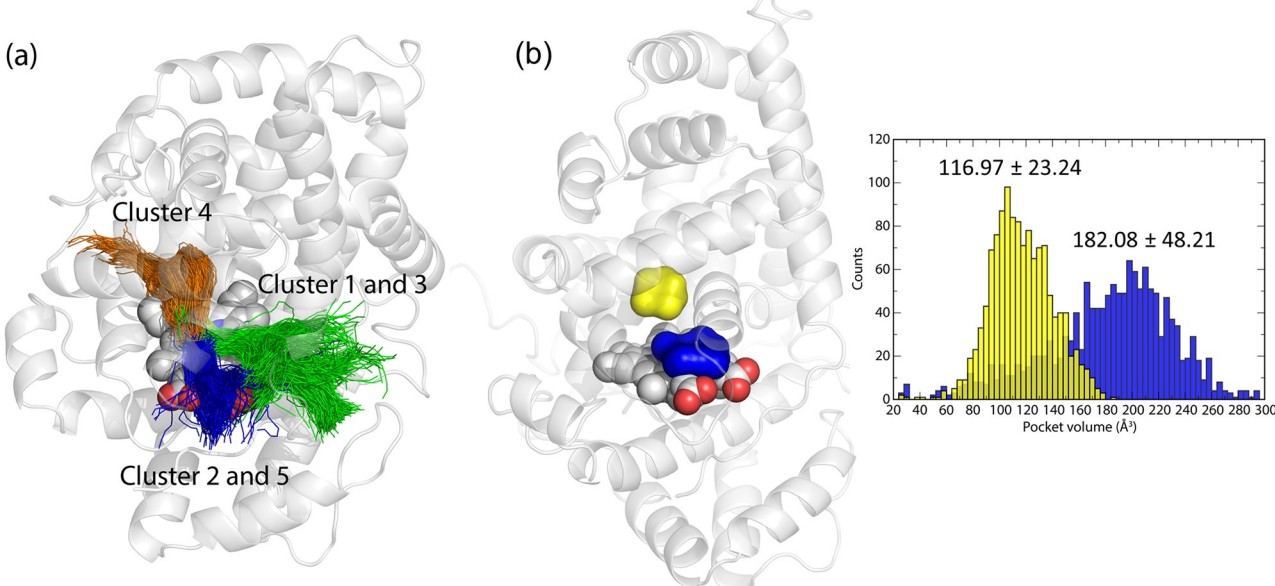

**Fig 4. Dynamical tunnel mapping and pocket identification in the LcpK30 open structure without substrate.** (a) Dynamical tunnel calculations with CAVER Analyst 2 using MD simulations of LcpK30 in open conformation. First 5 dominant tunnel clusters were calculated on 1500 snapshots from MD simulations of the open state. Two superclusters are formed, colored in green (comprised of clusters 1 and 3) and blue (comprised of clusters 2 and 5). Clustering displays the center lines for all tunnels computed for all snapshots at once. Center lines are colored according to their related clusters. (b) Pockets detected closest to heme calculated on 1500 snapshots from MD simulations of LcpK30 in the open state.

Interestingly, despite removing crystal waters from the crystal structure before the simulation, we observed a water network occupying the tunnels and both hydrophobic pockets above heme. Shortly after the relaxation, several buried protein sites, including the active site, were solvated, which was confirmed by obtaining a similar water occupancy from MD simulations as in the crystal structure. Moreover, we observed increased hydration of the tunnels in the open state compared to poorly hydrated tunnels in the closed state, again suggesting that these tunnels are more feasible in the open structure. This was also supported by the radial distribution function (RDF) of water oxygen atoms around the heme-bound $O_2$ indicating that the probability of water molecules approaching into the hydrophobic active site of LcpK30 is higher in the open conformation (S6 Fig).

Taken together, these results suggested an easier binding of larger substrates into the open compared to the closed conformation and indicated the existence of hydrophobic tunnels that provide access of the polymeric substrate to the active site. Next, we used docking of a polyisoprene substrate model to the open state structure to gain an understanding of the binding mode of the substrate to LcpK30.

## Molecular docking of the *cis*-1,4-polyisoprene substrate into LcpK30

Computational modelling of the interaction between flexible oligomeric ligands and their protein receptors is challenging. Dynamic modelling would be the best approach to account for the flexibility and obtain robust results [21, 22]. Modelling of the interaction between a polyisoprene-type substrate and LcpK30 has been previously reported in an elegant study by Zhang and Liu, in which they carried out docking of a short oligomer with four repeating units into the binding site near the heme cofactor containing a bound dioxygen [16]. In their

**Oligomeric substrate C$_{50}$H$_{82}$**

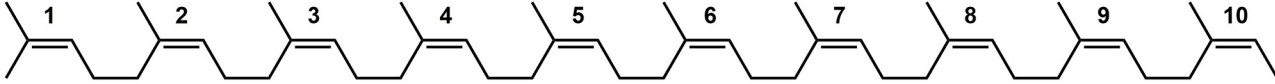

Fig 5. Substrate model with 10 repeating isoprene units (C$_{50}$H$_{82}$) used in this work.

work, the authors primarily focused on the QM/MM calculations of the LcpK30 catalytic mechanism featuring the oxidative cleavage of the C = C bond. However, such a short substrate model does not provide the full information about the interactions occurring further away from the active site, for example within the tunnels and hydrophobic pockets, that could additionally influence its binding and the catalysis outcome. On the other hand, the natural substrate of LcpK30 is a long polymer consisting of more than 10,000 repeating isoprene units, which would be too complex for standard docking into protein [23]. The major oligomers produced in the reaction with LcpK30 typically range from C20 to C35 (4 to 7 repeating isoprene units). Therefore, we reasoned that a substrate model with 10 repeating isoprene units (C$_{50}$H$_{82}$, Fig 5) was sufficiently large to provide valuable insights into the interaction of the polymer substrate with the enzyme. In an initial approach, we used docking to get an approximate indication of the binding of this substrate model to LcpK30, which we further refined with molecular dynamics simulations (see next section).

Our docking calculations were carried out in the GOLD software, using two different scoring functions, ChemPLP and ChemScore, which differ in the calculation of the fitness scores (see Methods). These scoring functions calculate docking scores by employing different empirical energy terms, such as van der Waals (VDW) energy, electrostatics, hydrogen bond, desolvation, entropy and hydrophobicity, to yield a dimensionless fitness score, which is optimised for ligand binding site prediction over binding affinity prediction. Hence, the fitness score provides an indication of the quality of the docking position. A higher (positive) fitness score indicates a higher probability that a ligand will bind to a protein in the given docking position, whilst a negative score indicates poor binding [24]. The crystal structure of LcpK30 in its open conformation was used as the protein receptor, where a dioxygen molecule was manually coordinated to the Fe(II) in heme. The binding site included all atoms within 10 Å of radius from the central iron atom in heme, which also contained the residues reported to be part of the active site of LcpK30 (Arg164, Lys167, Thr168 and His198) [12]. In the initial docking approach, the substrate was fully flexible while the protein was kept rigid. However, by using this rigid docking approach, we found that even the highest ranked poses had a very low, or even negative fitness score, indicating that significant steric clashes occurred between protein and substrate (S7 Fig and S2 Table). Therefore this docking approach was unsuitable to accurately describe the interaction between LcpK30 and the polyisoprene substrate.

To improve docking quality, induced fit (flexible) docking was carried out, accounting for flexibility of the receptor by allowing a full torsional flexibility of 10 amino acid sidechains in the active site of LcpK30 (see Methods for details and Table 2 for docking results). Increasing the size of the defined binding site from 10 Å to 15 Å was tested but was not suitable; although higher fitness scores were obtained, the oligomeric substrate tended to dock preferentially at the surface of the protein, without interaction with the heme and the active site of LcpK30 (S3 Table and S8 Fig). The 10 structures obtained using induced fit docking with each of the ChemPLP and ChemScore protocols were significantly different from those obtained with rigid docking and showed higher fitness scores. To obtain further information on their

orientation for catalysis, we analyzed the proximity of the C = C bond to the distal oxygen and the polymer conformation based on its span, its chain terminus position, and the position of the cleaved double bond (Table 2). In most docked poses, the distance between the C = C bond and the distal oxygen was below 5 Å, indicating that these structures could indeed represent pre-reactive states. Some exceptions were observed (ChemPLP docking pose rank 7; ChemScore docking poses 6 and 8), where the substrate was further away from the heme.

Overall, we found two prominent docked conformations of the oligomeric substrate, which we characterized as extended and folded. An extended conformation was defined as a structure in which one terminus of the oligomeric substrate model occupied the hydrophobic cavity within the protein interior near to heme (consistent with pocket 3 in our previous finding on the identification of tunnels and pockets, Fig 2b), interacting with the residues above the heme at a distal site, whilst the other terminus was exposed to solvent at the surface of the protein. A folded conformation was defined where both termini were found towards the surface of the protein (S9 Fig). In both these conformations, a C = C bond was found in the proximity of the distal oxygen, thus increasing the possibility of the reaction. The poses that were further away from the heme had an extended-like conformation. Interestingly, the higher docking scores using both ChemPLP and ChemScore functions were obtained with folded conformations. This probably occurred due to reduced steric clashes between atoms of neighboring residues

**Table 2. Structural features of the best docking poses obtained from induced fit docking within GOLD.** Docking solutions are ranked based on the fitness score from highest to lowest.

| Rank | Fitness Score | C = C closest to Fe−O$_2$ complex[a] | Distance of C = C to the oxygen atom (Å)[b] | | Ligand conformation |
|---|---|---|---|---|---|
| | | | -H$_2$C-C = | = C-CH$_3$ | |
| **ChemPLP docking protocol** | | | | | |
| 1 | 78.94 | 6 | 5 | 5.3 | Folded |
| 2 | 67.25 | 5 | 5.2 | 4 | Folded |
| 3 | 64.8 | 3 | 4.3 | 3.5 | Extended |
| 4 | 40.51 | 7 | 5.9 | 5.1 | Folded |
| 5 | 29.74 | 9 | 3.8 | 2.5 | Extended |
| 6 | 21.37 | 8 | 2.2 | 2.6 | Extended |
| 7 | 19.99 | 9 | 5.9 | 7 | Extended |
| 8 | -13.14 | 3 | 2.6 | 2.4 | Extended |
| 9 | -42.16 | 4 | 5 | 5.4 | Folded |
| 10 | -60.12 | 5 | 3.2 | 2.9 | Extended |
| **ChemScore docking protocol** | | | | | |
| 1 | 32.65 | 6 | 5.8 | 4.5 | Folded |
| 2 | 29.29 | 8 | 3.3 | 3.8 | Extended |
| 3 | 18.16 | 4 | 4 | 3.1 | Extended |
| 4 | 13.57 | 3 | 3 | 2.9 | Extended |
| 5 | 12.1 | 3 | 2.9 | 2.6 | Extended |
| 6 | 5.8 | 7 | 8.3 | 9 | Extended |
| 7 | 5.44 | 8 | 3.4 | 3.1 | Extended |
| 8 | -2.2 | 1 | 7.8 | 7.2 | Extended |
| 9 | -3.92 | 3 | 3.7 | 3.3 | Folded |
| 10 | -6.98 | 8 | 4.1 | 3.9 | Extended |

[a] the first C = C unit number was counted from the end terminal with two methyl group R-C = C-(CH$_3$)$_2$ −see Fig 5.

[b] the distance was measured from each C atom in the double bond to the distal O atom of the Fe-O$_2$ complex.

in the protein structure, which increased the fitness scoring function for the folded conformation.

PLIP analysis of the substrate-LcpK30 contacts revealed mostly hydrophobic interactions. Three residues, Lys167, Leu171 and Ile396 were involved in interactions with all docked conformations, with Ala159 involved in interactions with most docked conformations obtained with both ChemPLP and ChemScore (S4 and S5 Tables and S10 Fig). Residue Lys167 has previously been reported to serve as a gating mechanism that opens and closes the entrance to the hydrophobic channel of the enzyme active site [10]. Our docking results infer that this residue might also play an important role in stabilizing protein-substrate complexes. Furthermore, only the extended docked conformations interacted with the Glu148 side chain, located further inside the hydrophobic channel above the heme and previously suggested to play a function in fine-tuning the active pocket to accommodate the substrate for reaction [16]. Earlier work on mutation of Glu148 to alanine, histidine, and glutamine revealed that the glutamate residue affected the specific activity of the enzyme but showed no significant differences in biochemical and biophysical properties, as well as no change in oligoisoprenoid product distribution [10]. Given this, we hypothesize that the extended substrate conformation is more representative of a catalytically relevant pose of the substrate within LcpK30. However, it cannot be excluded that the cleavage occurs with the substrate in the alternative, folded conformation, given the overall higher docking scores and flexibility of the system.

Molecular dynamics simulation of LcpK30 in the presence of *cis*-1,4-polyisoprene.

We further explored the influence of substrate binding on the conformation of LcpK30 by performing MD simulations of the enzyme in the presence of the docked oligomeric substrate $C_{50}H_{82}$. We carried out simulations for all top 10 ranked solutions obtained with each of the ChemPLP and ChemScore fitness functions (20 poses in total).

Compared to the case without the substrate, relatively larger RMSD values of the enzyme C-alpha atoms were observed in the presence of the substrate, sometimes even above 2 Å, especially in the case of the poses in the extended conformation that bind deeper within the protein interior, such as docking solutions 10 and 3 from ChemPLP and ChemScore, respectively (Fig 6). From the RMSF calculations of the C-alpha atoms during the simulation of LcpK30 with the substrate bound, we observed overall very similar fluctuations as in the case of the enzyme without the substrate, which could indicate that these are characteristic for the protein in solution irrespective of whether substrate is present or not. However, when comparing the high RMSD docking solutions with the structure without the substrate, we characterized an increased fluctuation of protein residues 145–154 from a short helix interacting with the substrate bound in the hydrophobic pocket near the heme cofactor. Moreover, larger fluctuations were also observed for the gateway loop residues 155–160 and the helix E residues 160–176, containing the flexible Lys167 (see also S11 and S12 Figs for a visualization of RMSF values on the backbone of LcpK30).

The PCA analysis combining MD snapshots of the enzyme with and without substrate showed that compared to the empty enzyme, the presence of substrate slightly changed the conformation of the enzyme, irrespective of the extended or folded substrate binding mode (S13 Fig). The bound and unbound states were not separated along only one principal component, instead displacements along both PC1 and PC2 were observed. Normal mode displacements along the PC1 included the loop 155–160, helix E 160–176 and C-terminal helix Z2 297–314, whilst the displacement along the PC2 mostly localised at the C-terminal loop residues 392–397. As observed from the RMSD analysis, in some cases where the substrate binds deeper in the active site, the conformation of the enzyme bound to substrate was significantly different from the empty enzyme (for example in the case of extended substrate conformations 10 from ChemPLP and 3 from ChemScore). Furthermore, significant structural changes of the

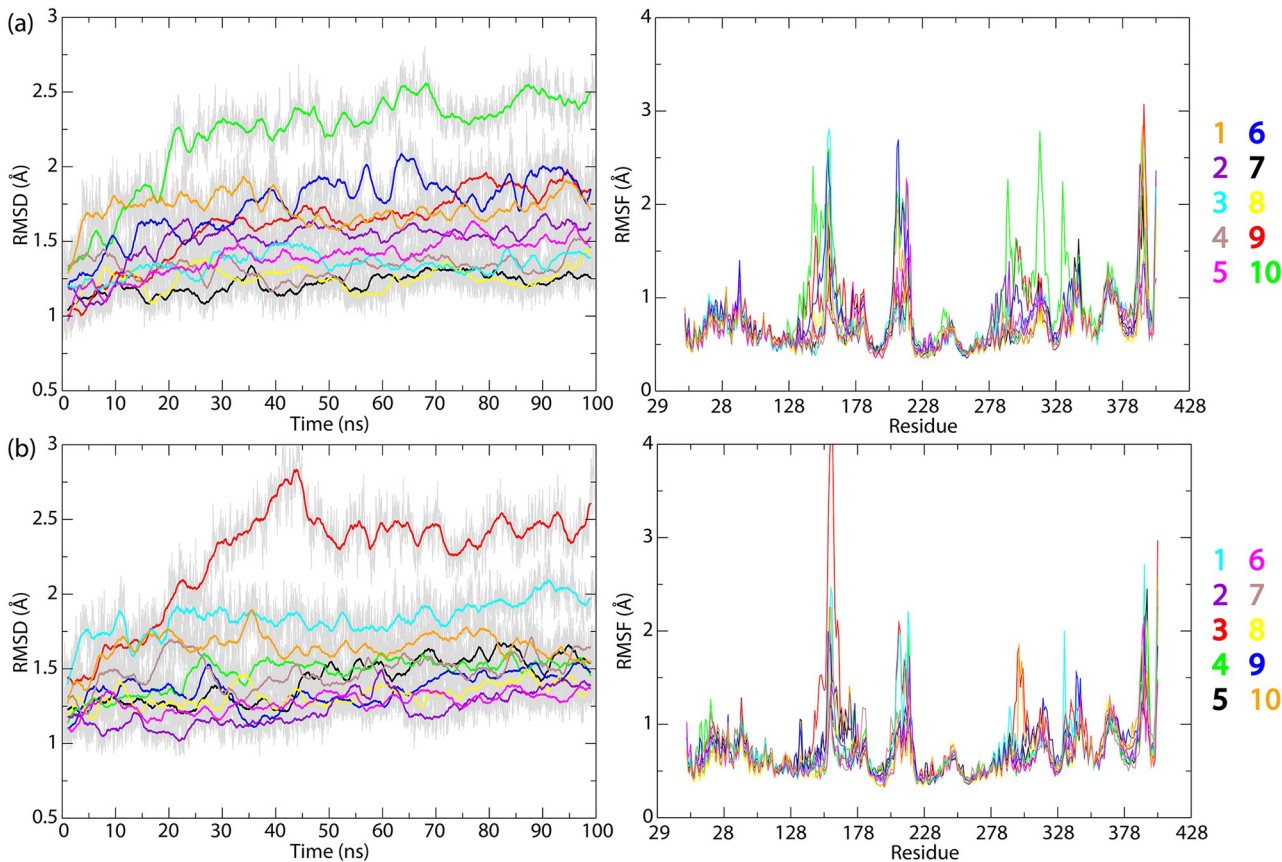

**Fig 6. RMSD and RMSF profiles for LcpK30 in the open state with *cis*-1,4-polyisoprene ($C_{50}H_{82}$) substrate model bound.** C-alpha RMSD (left) and RMSF (right) were calculated during 100 ns MD simulations, starting from 10 docking poses obtained with (a) ChemPLP and (b) ChemScore. All values are calculated from the reference X-ray structure of the enzyme. Residues 29–49 from N-terminus were omitted from the analysis to minimize the noise.

enzyme, especially C-terminus loop 392–397, were observed in case of simulations with the folded substrate conformations 1 from both ChemPLP and ChemScore.

The effects of the bound substrate on the tunnels leading to the active site were investigated next, by calculating the dynamical tunnel clusters within the complex between LcpK30 and *cis*-1,4-polyisoprene in extended and folded conformations, originating from both ChemPLP and ChemScore poses. The calculated features of the first 10 dominant clusters demonstrated that although throughputs were relatively high, and the priority of the individual tunnel clusters was very broad ranging from 9–44%, these tunnels had overall similar average and maximum bottleneck radius as in the case of LcpK30 in the closed-like state and without substrate bound (Fig 7 and Table 3, compare with Fig 4 and Table 1). However, the spatial distribution of tunnel clusters was similar to the open conformation without substrate, with two clearly separated superclusters due to the presence of the salt bridge between propionates of the heme cofactor and Lys167. All these results indicated that although mutual conformational changes of the enzyme and the substrate occurred (as supported by the PCA analysis) which influenced the throughput of the tunnels, a bound polyisoprene substrate strongly interacted with the enzyme within the characterized tunnels, which caused the narrowing of the bottlenecks.

Following the protein dynamics, we investigated the conformations of the substrate bound to the enzyme during MD simulations, to gain more insight into the stability of the complex

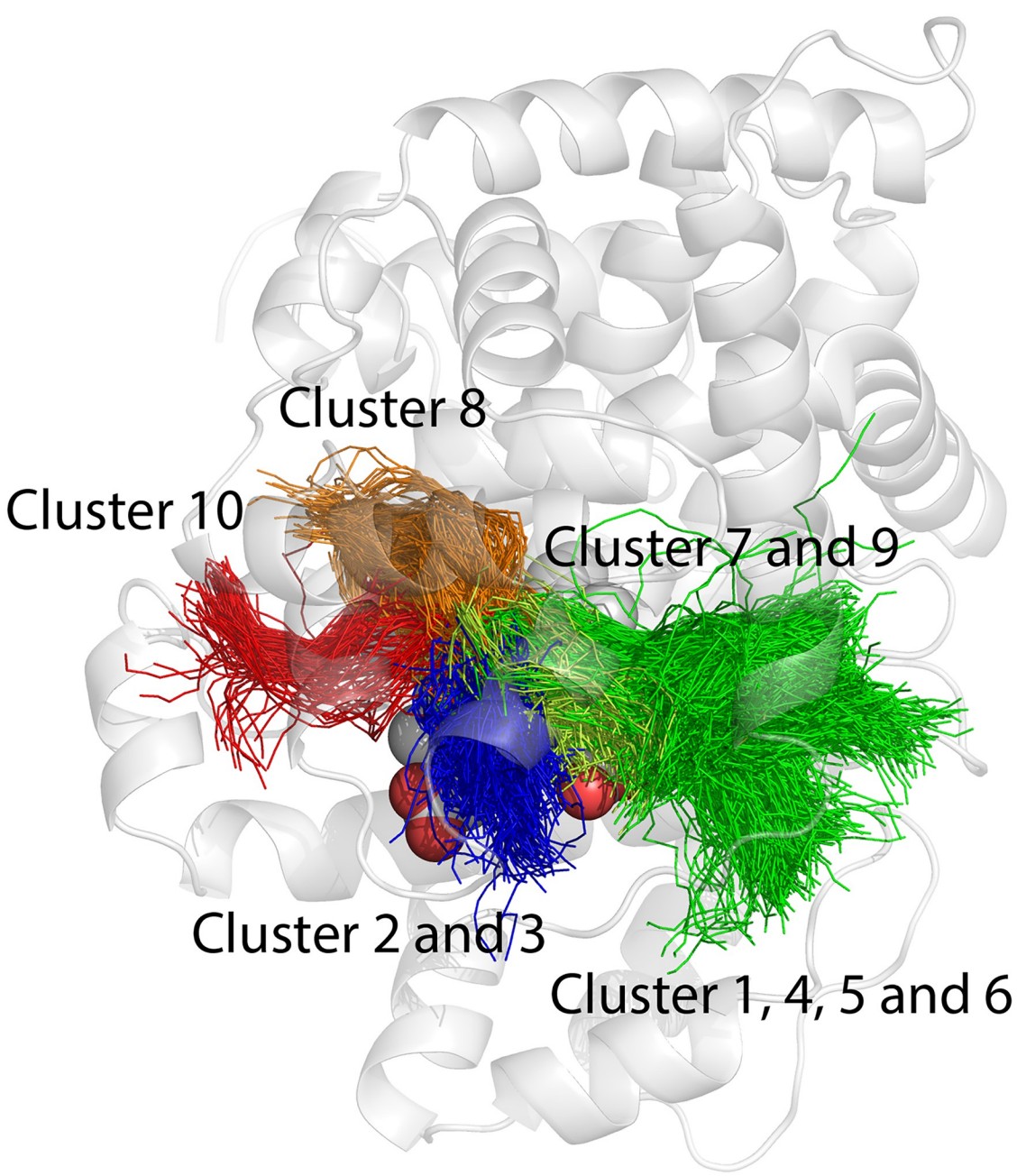

**Fig 7. Dynamical tunnel mapping and pocket identification in the LcpK30 open structure with bound substrate.** First 10 dominant tunnel clusters were calculated on 1000 snapshots from MD simulations of the open state LcpK30 with *cis*-1,4-polyisoprene substrate bound, using CAVER Analyst 2.

initially obtained with docking. While we observed minor fluctuations of the polymeric chain that binds deeper into the protein and tightly interacts with the heme cofactor, no significant conformational changes of the substrate occurred during 100 ns simulations, compared to the initial structures obtained with docking. Solvent-exposed regions experienced more flexibility (Fig 8), which could indicate that the interactions with the enzyme position and stabilize the flexible substrate in a suitable conformation for the reaction with the heme-bound oxygen.

**Table 3. Features of first 10 dominant tunnel clusters in LcpK30 bound to substrate.**

| ID | Average bottleneck radius (Å) | Std. dev. | Maximum bottleneck radius (Å) | Average length (Å) | Std. dev. | Average curvature | Std. dev. | Average throughput | Std. dev. |
|---|---|---|---|---|---|---|---|---|---|
| 1.08 | 0.08 | 1.21 | 16.05 | 4.28 | 1.49 | 0.28 | 0.67 | 0.11 | 1.08 |
| 1.05 | 0.08 | 1.21 | 11.90 | 3.17 | 1.39 | 0.20 | 0.66 | 0.12 | 1.05 |
| 1.09 | 0.07 | 1.21 | 6.73 | 3.11 | 1.29 | 0.21 | 0.80 | 0.07 | 1.09 |
| 1.06 | 0.08 | 1.21 | 25.88 | 3.73 | 1.38 | 0.13 | 0.49 | 0.10 | 1.06 |
| 1.05 | 0.08 | 1.21 | 26.25 | 3.28 | 1.50 | 0.20 | 0.49 | 0.09 | 1.05 |
| 1.06 | 0.08 | 1.21 | 23.15 | 3.70 | 1.38 | 0.19 | 0.52 | 0.10 | 1.06 |
| 1.07 | 0.07 | 1.21 | 11.80 | 3.15 | 1.40 | 0.24 | 0.72 | 0.07 | 1.07 |
| 1.02 | 0.09 | 1.21 | 18.56 | 3.76 | 1.43 | 0.21 | 0.49 | 0.11 | 1.02 |
| 1.07 | 0.08 | 1.21 | 9.07 | 3.57 | 1.71 | 0.55 | 0.73 | 0.12 | 1.07 |
| 1.02 | 0.08 | 1.21 | 24.41 | 3.53 | 1.61 | 0.19 | 0.40 | 0.09 | 1.02 |

Calculations were performed on 1000 snapshots from MD simulations of open state LcpK30 with *cis*-1,4-polyisoprene substrate bound. The tunnels were calculated starting from central Fe atom of heme and excluding $O_2$ and substrate.

Overall, these results indicate that, for the substrate to bind in the hydrophobic pocket of the active site, subtle mutual structural rearrangements of the LcpK30 receptor, especially around the gateway and of the polymeric chains are crucial for the formation of the stable complex.

To investigate the substrate-enzyme interactions, we calculated close contacts occurring in MD simulations and extracted information about the specific residues that most frequently interact with different ligand conformations (S6 and S7 Tables). Based on the interaction patterns, the two main types of bound conformations observed with docking (extended and folded) were confirmed. Similar to the docking results, some substrates were found in an extended-like conformation, without interactions with the hydrophobic cavity. The simulation confirmed that the distinctive difference between the extended and folded conformations arises from the fact that ligands in the extended conformation can reach the buried hydrophobic protein cavity, which is not the case for folded structures. Both conformations readily interacted with heme and $O_2$. The most frequent interactions and representative conformations of the substrate model bound in the active site of LcpK30 are shown in Fig 9.

Analyzing the frequent close contacts with the protein and the binding features of the extended and folded conformations of the *cis*-1,4-polyisoprene substrate bound to LcpK30 during MD simulations, we observed that both conformations readily interacted with the two previously found putative protein tunnels. The main differences between the two binding modes arose from the fact that only the substrate in the extended conformation could reach the buried hydrophobic cavity, which allowed it to favorably interact with the hydrophobic residues in the protein interior and to position a double bond near the heme-bound oxygen. Interestingly, we also found extended conformations that bound even deeper inside the protein, allowing the substrate termini to interact with the narrow tunnel cluster 4. In such conformations, one part of the polymeric chain was bound in the active site while the other was exposed to the surface interacting with either of the wider superclusters (blue and green in Fig 4, clusters 1, 2, 3 or 5). In contrast, the substrate in the folded conformation did not reach the hydrophobic pocket but rather coiled around Lys167 and the flexible C-terminus loop comprised of residues 392–397, with a central double bond positioned near the heme active site in an orientation favorable for catalysis.

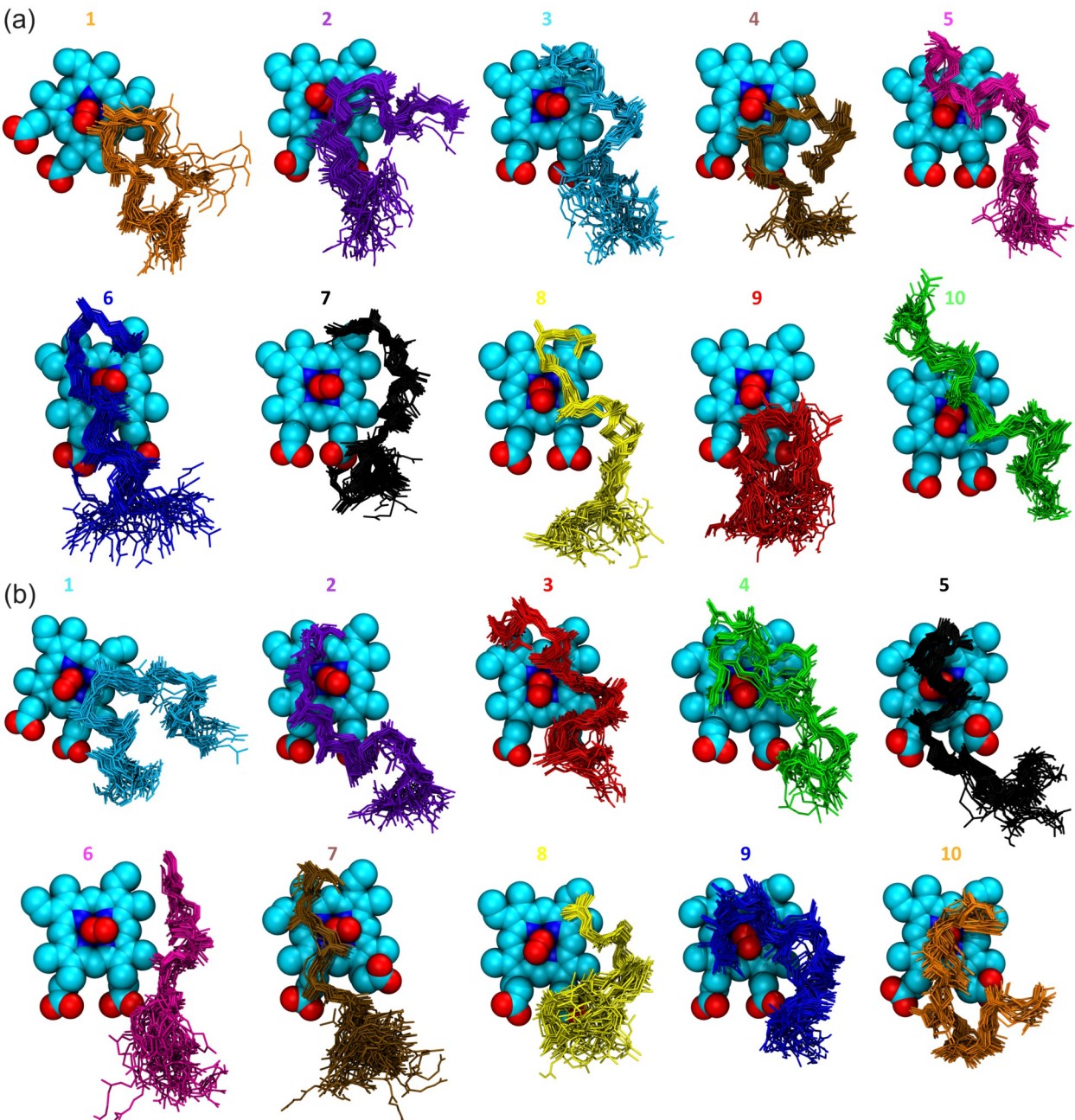

**Fig 8. Ensemble of 50 superimposed snapshots from 100 ns MD simulations of *cis*-1,4-polyisoprene near heme, obtained starting from the 20 top ranked docking poses.** a) Snapshots obtained from ChemPLP fitness function; b) Snapshots obtained from ChemScore fitness function. Trajectory smoothing window size was 5. Protein and hydrogens were omitted for the sake of clarity.

To expand our investigation, we explored dominant energy factors that drive the *cis*-1,4-polyisoprene binding to LcpK30, by calculating van der Waals and electrostatic energy terms with linear interaction energy [25] on structures of the complex from MD simulations of all docking solutions from ChemPLP and ChemScore (Fig 10). Overall, we found that $E_{vdW}$ dominates the total interaction energy with 94% contribution over $E_{ele}$, which contributes only

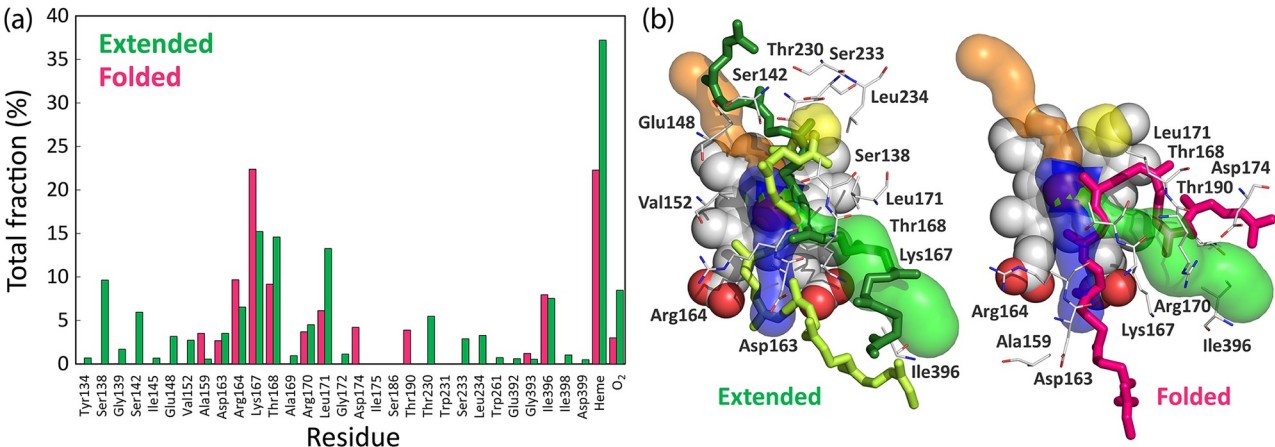

**Fig 9. Residues that frequently interact with the substrate during MD simulations.** (a) Close contacts between frequently occurring bound conformations of the substrate (extended and folded) and LcpK30, calculated from 100 ns MD simulations of 20 docking poses obtained with ChemPLP and ChemScore fitness function. The profile was constructed by averaging total fractions obtained for similar conformations. Conformations were clustered based on the contacts and the features from the visual inspection; (b) Representative snapshots extracted from MD simulations showing the main extended (ChemPLP pose 10 and ChemScore pose 5 shown as dark green and light green licorice, respectively) and folded (ChemScore pose 1 shown as magenta licorice) conformations and protein residues (shown as lines) that interact with the *cis*-1,4-polyisoprene substrate model bound to LcpK30 near heme (sphere representation). The three tunnels and the hydrophobic cavity are shown as blue, green, orange and yellow surface, respectively. The rest of the LcpK30 and hydrogen atoms are omitted for clarity.

6%. Furthermore, the binding energy was calculated with the MM/GBSA method [26] using snapshots from MD simulations and treating the solvation implicitly. Interestingly, the calculated binding energies show similar trends as observed for the total interaction energies. Whilst most docking solutions, irrespective of the conformation, have similar binding energy, structures 9 and 10 from the ChemPLP fitness score have significantly higher or lower binding energy, respectively. We further analyzed the relationship between interaction energies, and

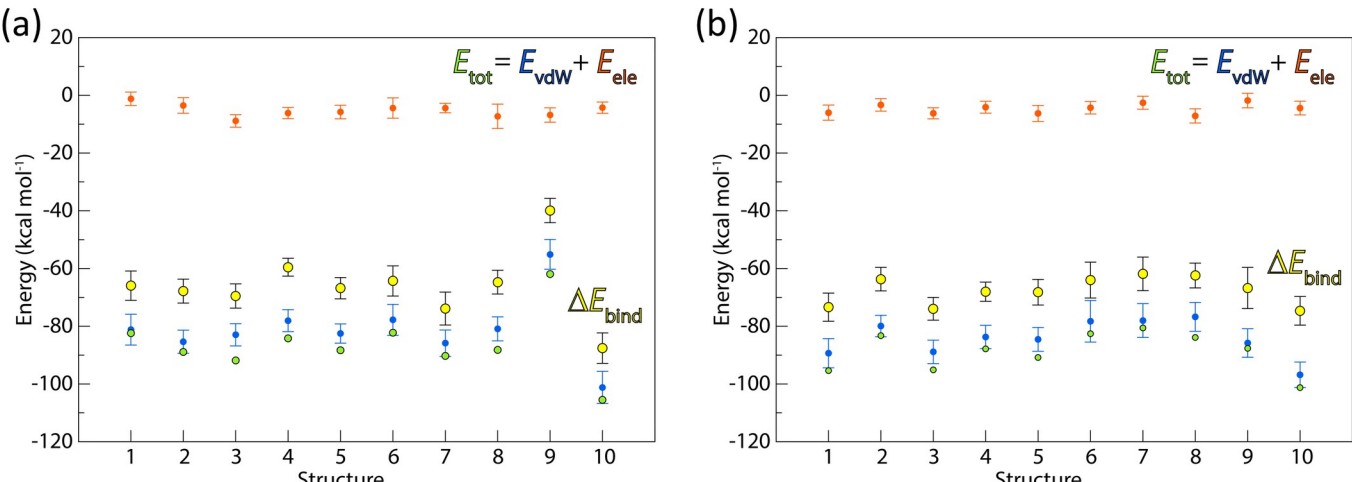

**Fig 10. Linear interaction energy analysis between the ligand and Lcp$_{K30}$.** Calculations were performed on 100 ns MD simulations of 10 docking poses obtained with (a) ChemPLP and (b) ChemScore fitness function. The $E_{ele}$ is the electrostatic energy (orange circles) and $E_{vdW}$ is van der Waals energy (blue circles). The total energy $E_{tot}$ (green circles) is calculated as a sum of $E_{ele}$ and $E_{vdW}$. The binding energy $\Delta E_{bind}$ (yellow circles) was calculated on 50 snapshots from MD simulations with the MM/GBSA method.

therefore binding affinities, and the total number of close contacts between the substrate and LcpK30 during MD simulations. We found that the binding affinity decreases when there are too few or too many close contacts. For example, the conformer with the most favorable binding affinity had 9 frequent contacts with the protein residues, while the conformer with the least favorable binding had only two contacts less. However, the conformation that achieved the least interactions with the protein (5 close contacts) had less favorable binding energy than the structure that had the most interactions (15 contacts). Moreover, we observed that the substrates in the extended conformation typically achieved 9 or more frequent contacts with the protein residues, while this number is less than 9 close contacts in the case of substrates that bind in the folded conformation (S6 and S7 Tables). It is important to mention that the calculated binding energies do not correlate with the docking fitness scores, which highlights the importance of the flexibility of the system and the advantage of exploring different bound conformations of a ligand with molecular simulations.

The cleavage sites in the *cis*-1,4-polyisoprene substrate model were predicted by constructing the histograms of distances calculated between distal oxygen atom from heme-$O_2$ cofactor and each of the ten C = C double bonds of the substrate during MD simulations. The near-attack-conformations of the substrate, in which the C = C bonds are frequently found around or below 4 Å from the $O_2$, were considered suitable for the oxidative cleavage by LcpK30 (S14 and S15 Figs). We used this information to construct the oxidative cleavage frequency profile by assigning the total number of potential cleavage events to a specific C = C bond based on their distance from the reactive $O_2$. Fig 11 shows that the central C = C double bonds (numbered 5 and 6, see Fig 5) of the substrate were more frequently found in the near vicinity of heme-$O_2$ and therefore more prone to oxidative cleavage into two oligomers of 4 and 5 isoprene units. This can partially explain the previously reported experimentally observed distribution of oligomers produced by LcpK30 which shows a preference for products containing 4–7 intact isoprene units [1, 8, 15]). Furthermore, 4 out of the 6 folded conformations of the substrate had central C = C bonds in a favorable position for cleavage. On the other hand, the oxidative cleavage of the substrates in the extended conformation seemed to be more diverse and preferably occurred at the C = C bonds 3, 4 and 8 (S14 and S15 Figs).

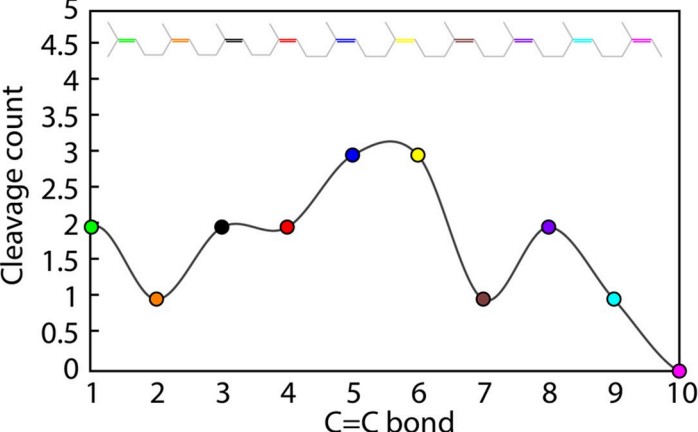

**Fig 11. Cleavage frequency profile.** Cleavage frequency profile constructed from distance histograms indicating a total count of each of the ten C = C double bonds (coloured differently) from the *cis*-1,4-polyisoprene substrate that are found in the near vicinity (below 4 Å) of the distal oxygen atom in the heme-bound $O_2$ molecule. The profile is calculated from 100 ns MD simulations of LcpK30 in the presence of the substrate for all 20 docking poses obtained with ChemPLP and ChemScore fitness functions.

## Conclusion

Knowledge about pathways that lead from the surface to the active site is important to understand enzymatic catalysis. Frequently, the residues comprising these pathways actively transport substrates and products between the active site and bulk solvent, which is particularly important for enzymes with buried active sites [27–29]. In this work, docking and molecular dynamic simulations provided a deeper understanding of the interaction between the *cis*-1,4-polyisoprene and the enzyme latex clearing protein, including a detailed characterization of the tunnels and hydrophobic pockets in LcpK30. Using the CAVER tunnel identification and the automated fpocket tools within the static LcpK30 structure, as well as MD simulation analyses, we characterized potential entry and exit pathways, as well as hydrophobic pockets for the binding of oligomeric *cis*-1,4-polyisoprene chains within the enzyme. A range of residues interacting with the substrate were identified and comprised Ser138, Ser142, Glu148, Val152, Arg164, Lys167, Thr168, Leu171, Thr230, Ser233 and Leu234. We suggest that these residues represent suitable targets for mutagenesis aimed at improving enzymatic degradation rates and/or substrate and oligoisoprenoid product range of LcpK30.

Computational docking of a large substrate model with 10 repeating isoprene units ($C_{50}H_{82}$), using two different scoring functions within the GOLD program, revealed two main bound conformations: extended and folded. Residues Lys167, Leu171, Ile396 and Ala159 were involved in the protein-substrate interactions in most of the docking solutions, and we suggest they play an important role in stabilizing protein-substrate complexes. Only the extended substrate conformations interacted with residue Glu148, which was previously shown to affect enzyme activity. Therefore, we suggest that the extended conformation is more representative of a catalytically relevant substrate pose within the enzyme. Further analysis using MD simulations revealed additional close contacts between the substrate in different conformations and the enzyme. This analysis also confirmed that hydrophobic (van der Waals) interactions were mostly responsible for mutual subtle conformational changes of both LcpK30 and the polymeric substrate upon binding. Furthermore, an oxidative cleavage frequency profile was constructed.

Taken together, the tunnel and pocket predictions, docking results and MD simulations suggest that the substrate enters *via* a gateway comprised of flexible loop 155–160 that extends to the helix 160–176 and the C-terminus residues 392–397 which could also serve as the potential exit pathway for the oligoisoprenoid products (Fig 12). A hydrophobic pocket exists above the heme, where the substrate binds, and which is likely to determine the size distribution of the oligomers. An optimal number of hydrophobic interactions favors better substrate binding, which suggests the substrate needs to bind deeper into the protein cavity for the reaction to occur and may explain the oligomer sizes obtained during cleavage. We hypothesize that the most favorable substrate binding occurs in an extended conformation, with one terminus within the hydrophobic pocket and the other terminus at the surface of the protein. A second binding mode with the substrate in a folded conformation also occurred, with a central double bond in proximity to the heme-bound oxygen, leading to a preference for oligomers with 5 to 6 C = C bonds, which can partially explain previously reported experimental results showing cleavage into mainly 4–7 intact isoprene units.

Whilst experimental evidence is required to confirm the substrate binding modes, the results presented here provide valuable hypotheses on how latex clearing proteins can incorporate and cleave oligomeric substrates and suggest suitable targets for mutagenesis aimed at improving enzymatic degradation rates. Furthermore, the thorough computational investigation on the structure, interaction, and catalytic mechanism of LcpK30 discussed in this work will provide deeper insight into the understanding of enzymatic catalysis with other large, flexible and hydrophobic macromolecules.

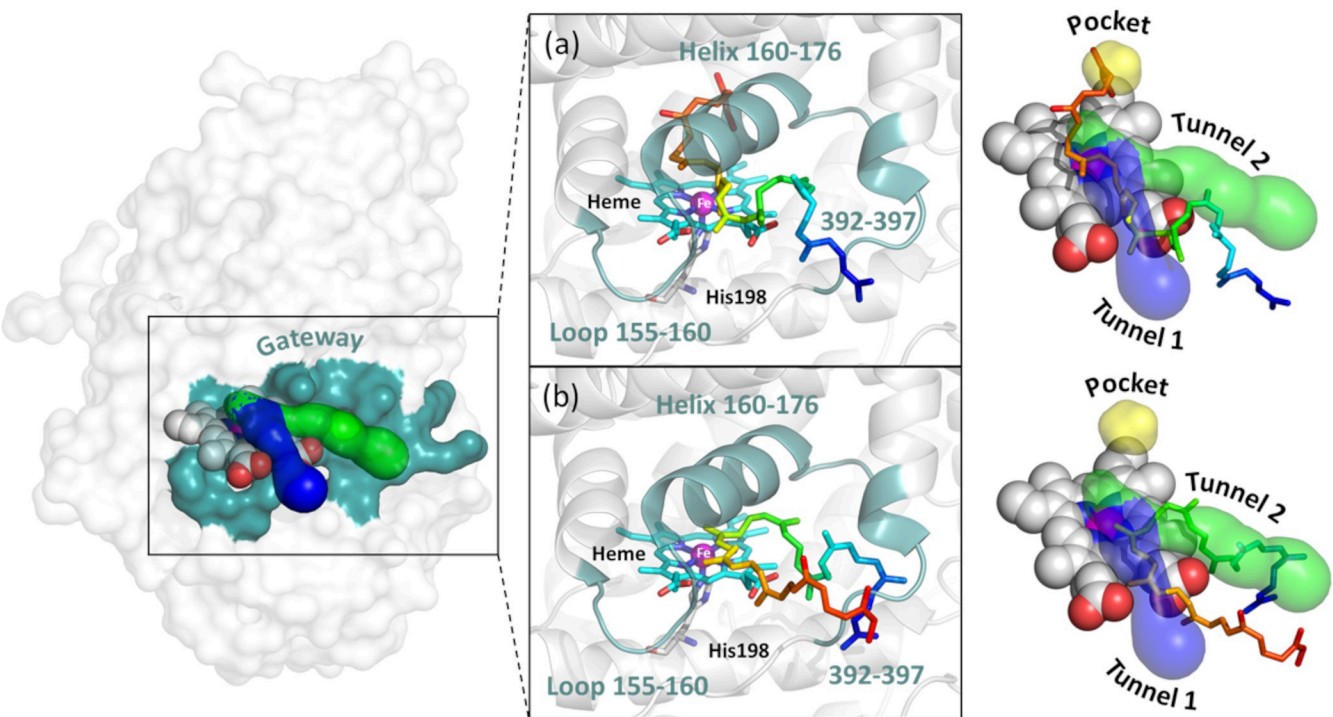

**Fig 12. Proposed gateway for the substrate entry into LcpK30.** The gateway residues were proposed based on the results from static and dynamic modelling of LcpK30 with *cis*-1,4-polyisoprene ligand. The representative snapshots of (a) extended and (b) folded conformation of *cis*-1,4-polyisoprene in complex with LcpK30 highlighting their interaction with the putative tunnels and the hydrophobic protein cavity. The $O_2$ bound on heme was omitted for the sake of clarity.

## Computational methods

### Protein structure preparation

The high-resolution crystal structures of the LcpK30 enzyme in the open and closed conformations were obtained from the RCSB Protein Data Bank (PDB ID: 5O1L for open and 5O1M for closed) [10]. Although the LcpK30 is a biologically active dimer, only one monomer (chain A) was retained for all calculations and modelling because the interface between the two monomers is relatively far from the active site. The corresponding heme cofactor was also retained, while all other co-crystallized species including imidazole, 2,3-butanediol and 1,2-ethanediol molecule and waters were removed prior to all calculations.

### Protein tunnel calculations

Putative tunnels were calculated with the CAVER-Pymol plugin 3.0.3 [19]. The hydrogen atoms were added to the protein structure using Chimera 1.16, considering the H-bonds network [30]. The initial starting point for the tunnel search was specified by the central iron atom of the heme prosthetic group. The search was performed using a minimum probe radius of 0.9 Å. Default parameters for the maximum distance of 3 Å, desired radius of 5 Å, shell depth of 4 Å, and shell radius of 3 Å were used [19]. The clustering of tunnels was performed by the average-link hierarchical Murtagh algorithm based on the calculated matrix of pairwise tunnel distances. The clustering threshold was set to a default value of 3.5. Each atom of the protein was approximated by 12 spheres. Calculations of tunnels on snapshots from MD simulations was carried out in CAVER Analyst 2 using default parameters [31].

## Protein pocket calculations

Simple pocket detection was performed with the open source protein pocket (cavity) detection algorithm based on Voronoi tessellation known as fpocket [32]. The detection of pockets on MD trajectories of LcpK30 was carried out with mdpocket considering protein and heme atoms only [33]. Default values of parameters to describe pockets were used. Namely, the minimum and the maximum radius that an alpha sphere might have in a binding pocket was 3.4 Å and 6.2 Å, respectively. The pairwise single linkage clustering method with the Euclidean distance measure was used for the clustering algorithm. Each pocket contained at least 15 alpha spheres and pockets volume were calculated using a Monte-Carlo algorithm.

## Protein-ligand docking

The GOLD program (version 2022.2.0) was used for the molecular docking protocols, with the default parameters as defined within the program [24, 34]. The protonation states of the titratable residues were assigned using the H++ server [35]. To simulate the catalytically active state of the protein, the imidazole molecule coordinated on the iron of heme was replaced with $O_2$ that was manually added. The 2D structure of the C50 *cis*-1,4-polyisoprene ligand model was constructed using ChemDraw 18.2 (Perkin Elmer, MA, USA) and the 3D structure with all hydrogen atoms included was obtained using Chem3D 18.2 (Perkin Elmer, MA, USA).

The ChemPLP and ChemScore fitness functions available in GOLD were used [34]. These scoring functions were selected because they both perform well for lipophilic binding sites and accurately model the steric complementarity between protein and ligand. All docking runs were performed with default parameters using both rigid and flexible docking. The binding site was defined as all protein residues whose atoms are within 10 or 15 Å from the central iron of heme. Ligand flexibility was enabled during all docking runs. To include flexibility of the active site, induced fit (flexible) docking was carried out, where the side chains of all residues that have been identified as important to the catalytic activity of LcpK30 (Glu148, Arg164, Lys167, Thr168) were specified as fully flexible [12]. Several other residues which were found in the pathway identified from the static tunnel calculations using CAVER (Ser142, Ile145, Val152, Leu171, Thr230 and Leu234) were also defined as flexible. These side chains were allowed to rotate freely during docking to vary over the range -180 to +180 rotatable torsion degree. The protein-ligand interaction was analyzed using Protein-Ligand Interaction Profiler (PLIP) program [36].

## Molecular dynamics simulations

The protonation states of the titratable residues at the physiological conditions were assigned using the H++ server [35]. The enzyme coordinates were taken from the crystal structure of the open (PDB ID: 5O1L) conformation of LcpK30, chain A [10]. The classical ff19SB [37] force field parameters were assigned to the standard residues, while the bonded model was used to describe the oxyheme cofactor and the coordinating His198. The cofactor was parametrized with the metal center parameter builder (MCPB) protocol [38]. To generate bonded force field parameters of the heme group with MCPB.py, we used a crystal structure of LcpK30 in the open state (PDB: 5O1L) [10] in which the central iron is coordinated by four equatorial nitrogen atoms from the protoporphyrin ring and one axial epsilon-nitrogen atom from the histidine sidechain of the enzyme. We manually included an oxygen molecule bound to the remaining iron site. We applied the B3LYP/6-31G(d) level of theory to carry out the geometry optimization for both the small and large models. Apart from the heme cofactor and $O_2$ molecule, the small model had the sidechain of the coordinating histidine, while the large model also included the histidine backbone. The force constants were calculated for the optimized

small model using the same method. Prior to the Merz-Kollman RESP partial charges calculation we performed minimization of the large model allowing the relaxation of hydrogen atoms and of $O_2$ to the optimal positions. The +2 charge was defined for the iron atom, and the total charge of metal site was -2, while the metal site had a possibility to adopt multiple electronic structures. We tested different spin states and concluded that the bond parameters are closer to the experimental crystal structure of oxymyoglobin in the singlet spin state. The ESP charges, and consequently RESP charges, were similar both in the case of low and high spin systems. We employed the Seminario method to obtain force field parameters using the ff19SB/GAFF force field and ChgModB to perform the RESP charge fitting. All QM calculations were carried out with Gaussian 16 program [39].

To explore conformational dynamics of the enzyme and the interactions of the enzyme with the ligands, we simulated LcpK30 in the absence and in the presence of polyisoprene ligand C50 *cis*-1,4-polyisoprene with 10 C = C double bonds. We used all of the top 10 ranked solutions obtained with each of the ChemPLP and ChemScore fitness functions in the GOLD software (20 poses in total). The GAFF force field was used to describe bonded parameters of the ligands while the partial charges were calculated using the AM1-BCC method [40, 41]. We solvated the systems with OPC [35] water using the truncated octahedron box with each atom of the solute at least 10 Å away from the edge of water surrounding the solute. All systems were neutralized by adding $Na^+$ counterions. The 12–6 set of parameters for monovalent atomic ions for the OPC water model were used [42]. All systems were minimized for a total of 10,000 steps, first 5,000 steps of the steepest descent method and the conjugate gradient for the remaining cycles. In the first relaxation step, the temperature of the system was gradually increased from the initial temperature of 0 to a target temperature of 300 K over 100 ps and kept constant for another 100 ps of the NVT simulation. We employed the Langevin thermostat with the collision frequency of 2 ps for the temperature control. In the second step of the relaxation, the system was propagated at 300 K over 100 ps of simulation time at a constant pressure (NPT), allowing the box to relax using the Berendsen barostat. The weak positional restraints (5 kcal mol$^{-1}$ Å$^{-2}$) were applied to the solute in the relaxation phase. This last step of relaxation included the removal of all restraints on the solute for 2 ns of simulation time at a constant pressure. Finally, for LcpK30 without the ligand, three independent production runs were carried out at 300 K and the constant pressure for at least 500 ns of simulation time. The production simulations of LcpK30 with the ligand were carried out for 100 ns each. All docking poses obtained from ChemPLP and ChemScore fitness functions were simulated. The Particle Mesh Ewald approach was used for treating long-range electrostatic interactions and the non-bonded cut-off was set to 10 Å. The SHAKE algorithm was used to constrain bonds involving hydrogen and a time step of 2 fs was used in all simulations. All MD simulations and the trajectory analysis were carried out in the AMBER20 program [43]. The visualization was performed in Pymol 2.5.4 [44] and VMD 1.9.3 [45].

## Supporting information

**S1 Fig. Crystal structure analysis.** Close view of the active site from the crystal structure of Lcp$_{K30}$ showing the helix 160–176 near heme in (a) open state (PDB: 5O1L) and (b) closed state (PDB: 5O1M).
(TIF)

**S2 Fig. Tunnel profiles for the dominant static tunnels.** Profiles were calculated using the Lcp$_{K30}$ crystal structure in the open state (PDB 5O1L) using the CAVER-Pymol plugin 3.0.3. (a) Variation of tunnel radius with length; (b) tunnel profile heatmaps; (c) distances between different atoms in the *cis*-1,4-polyisoprene chains, used to estimate maximum width of the

chain in an extended conformation. The distances were measured between the indicated carbon atoms that represent the bond length (blue) and close contact atoms (red). The addition of C-H distances is anticipated to add max. 1.1 Å.
(TIF)

**S3 Fig. RMSD and RMSF profiles for LcpK30 in the open state.** (a) C-alpha RMSD and (b) RMSF values were calculated from 3 independent 500 ns MD simulations (black, red and green) of $Lcp_{K30}$ from the reference X-ray structure in the open state. Due to increased flexibility of the N-terminus, residues 29–49 were omitted from the analysis to minimize the noise. The RMSF values are shown per residue as average from all 3 trajectories.
(TIF)

**S4 Fig. Backbone flexibility of open state LcpK30 from MD simulations.** (a) Experimental B-factor from the X-ray structure and (b) C-alpha RMSF (Å) values from MD simulations plotted on the backbone of $Lcp_{K30}$. Due to large fluctuations of the N-terminus, residues 29–49 were omitted from the analysis of the MD trajectories for the sake of clarity. The heme cofactor is represented in spheres (orange = Fe, white = C, red = O, blue = N). The protein structures are depicted with the B-factor putty representation where the backbone is displayed as a tube with a diameter correlated to the experimental B-factor from the X-ray structure (a) or RMSF from MD simulations (b). Thicker tube indicates higher flexibility. The structures are coloured with a continuous scale that ranges from blue to red to indicate the backbone mobility, where blue is low, and red is high backbone flexibility.
(TIF)

**S5 Fig. Dynamical tunnel mapping and pocket identification in the LcpK30 closed-like structure.** First 5 dominant tunnel clusters were calculated with CAVER Analyst 2 on 1500 snapshots from MD simulations of the closed-like state. Clustering displays the centre lines for all tunnels computed for all snapshots at once. Centre lines are coloured according to their related clusters. Clusters 1, 3, 4 and 5 form one narrow supercluster, colored in limon.
(TIF)

**S6 Fig. Dynamic of water molecules during simulations of open and closed-like states of LcpK30.** Radial distribution function (RDF) of water oxygen atoms from $O_2$ molecule bound to heme in (a) open and (b) closed LcpK30. The isosurface (isovalue = 100) showing the density of water oxygen atoms around the active site in (c) open and (d) closed LcpK30. Top view at the heme cofactor in the active site of LcpK30 showing water molecules as blue surface using the representative MD snapshots in (e) open and (f) closed LcpK30. The analysis was carried out on 6000 frames from 3 repeat MD simulations of open and closed-like LcpK30 states.
(TIF)

**S7 Fig. Superimposed conformations results from rigid docking of cis-1,4-polyisoprene C50H82 within LcpK30.** (a) Best 10 poses obtained using the ChemPLP scoring function; (b) best 10 poses obtained using ChemScore scoring function.
(TIF)

**S8 Fig. Effect of binding site size on docking.** Superimposed docking conformations results of $C_{50}H_{82}$ within a defined radius of (a) 10 Å and (b) 15 Å from the central iron atom in heme using ChemPLP flexible docking. Ligand is represented as yellow sticks, and the heme cofactor is represented in spheres (white = C, red = O, blue = N).
(TIF)

**S9 Fig. Docking conformations of cis-1,4-polyisoprene (C50H82) near heme.** Docking conformations of *cis*-1,4-polyisoprene ($C_{50}H_{82}$) near heme. Ranking is by the fitness score from highest to lowest. a) Poses obtained with the ChemPLP fitness function; b) poses obtained with the ChemScore fitness function. Docking solutions were calculated with induced fit docking using a 10 Å binding site. Enzyme and hydrogens were omitted for the sake of clarity. (TIF)

**S10 Fig. Protein-ligand interactions identified by PLIP analysis.** Protein-ligand interactions identified by PLIP analysis. Docked poses were obtained with ChemPLP (above) and Chem-Score (below). (a) docking solutions ranked 1, representative of folded conformations and (b) docking solutions ranked 3, representative of extended conformations. Docking solutions were calculated with induced fit docking using a 10 Å binding site and ranked based on the fitness score from highest to lowest (see Table 1). The hydrophobic interactions between the ligand and interacting residues are shown by dashed lines. The three residues that appeared in all interactions are underlined. The residue that interacted only with the extended conformation is coloured in blue. The red coloured residue (Ala159) appeared in all interactions with poses obtained using ChemScore and in most interactions with poses obtained using ChemPLP. (TIF)

**S11 Fig. C-alpha RMSF calculated during 100 ns MD simulations of LcpK30 with cis-1,4-polyisoprene bound, starting from 10 docking poses obtained with ChemPLP fitness function.** C-alpha RMSF calculated during 100 ns MD simulations of Lcp$_{K30}$ with *cis*-1,4-polyisoprene bound, starting from 10 docking poses obtained with ChemPLP fitness function. All values are calculated from the reference X-ray structure of the enzyme and projected on the average structure of the enzyme. Due to increased flexibility, residues 29–49 from N-terminus were omitted from the analysis to minimise the noise. The substrate is omitted for the sake of clarity. The protein structures are depicted with the putty representation where the backbone is displayed as a tube with a diameter correlated to the RMSF from MD simulations (thicker tube indicates higher RMSF). The structures are coloured with continuous scale that ranges from blue to red to indicate the backbone mobility, where blue is low and red is high backbone flexibility. (TIF)

**S12 Fig. C-alpha RMSF calculated during 100 ns MD simulations of LcpK30 with cis-1,4-polyisoprene bound, starting from 10 docking poses obtained with ChemScore fitness function.** C-alpha RMSF calculated during 100 ns MD simulations of LcpK30 with cis-1,4-polyisoprene bound, starting from 10 docking poses obtained with ChemScore fitness function. All values are calculated from the reference X-ray structure of the enzyme and projected on the average structure of the enzyme. Due to increased flexibility, residues 29–49 from N-terminus were omitted from the analysis to minimise the noise. The substrate is omitted for the sake of clarity. The protein structures are depicted with the putty representation where the backbone is displayed as a tube with a diameter correlated to the RMSF from MD simulations (thicker tube indicates higher RMSF). The structures are coloured with continuous scale that ranges from blue to red to indicate the backbone mobility, where blue is low and red is high backbone flexibility. (TIF)

**S13 Fig. Principal component analysis (PCA) carried out on structures from MD simulations of open and closed LcpK30 with and without substrate.** PCA was performed considering cartesian coordinates of protein backbone atoms (N, Cα, C and O). (a) MD snapshots

projected on the first two principal components. Blue and red circles represent MD snapshots belonging to open and closed-like states, respectively, orange circles represent MD snapshots belonging to the open state LcpK£0 in complex with the substrate. (b) Normal mode displacement vectors associated with the first two principal components showing only motions longer than 2 Å as porcupine in both directions. The backbone is coloured by the mobility where lower and higher flexibility is depicted in blue and red, respectively. Structures from MD simulations projected on the first two principal components considering open LcpK30 (grey) and LcpK30 in complex with substrate from (c) ChemPLP and (d) ChemScore docking poses. (TIF)

**S14 Fig. Probability distribution of distance between distal oxygen of O2 and each C = C double bond for docking poses obtained with ChemPLP.** Calculations were during 100 ns MD simulations of $Lcp_{K30}$ with *cis*-1,4-polyisoprene bound, starting from 10 docking poses obtained with ChemPLP fitness score. The cleaved bonds were the ones found closest to $O_2$. The color of the C = C double bond (see scheme at the bottom) corresponds to the color of the probability histogram.
(TIF)

**S15 Fig. Probability distribution of distance between distal oxygen of O2 and each C = C double bond for docking poses obtained with ChemScore.** Calculations were during 100 ns MD simulations of $Lcp_{K30}$ with *cis*-1,4-polyisoprene bound, starting from 10 docking poses obtained with ChemScore fitness score. The cleaved bonds were the ones found closest to $O_2$. The color of the C = C double bond (see scheme at the bottom) corresponds to the color of the probability histogram.
(TIF)

**S1 Table. The features of dominant tunnels calculated using the LcpK30 crystal structure in the open state (PDB 5O1L) using the CAVER-Pymol plugin 3.0.3.**
(PPTX)

**S2 Table. Docking solutions obtained from rigid docking within GOLD software.** Solutions are ranked based on the fitness score from highest to lowest.
(PPTX)

**S3 Table. Impact of the size of the defined protein binding site on the ChemPLP docking scores.** Docking solutions are obtained from flexible docking using the ChemPLP scoring function within the GOLD software. The docking was performed in a binding site defined as all atoms within 10 or 15 Å from the central iron atom in heme. The 10 docking solutions are ranked based on the fitness score from highest to lowest.
(PPTX)

**S4 Table. List of residues interacting with the docked cis-1,4-polyisoprene, extracted from PLIP analysis of 10 docking poses obtained using ChemPLP.** Docking solutions were ranked based on the fitness score from highest to lowest. Residues that interact with all / most poses are highlighted in grey and light grey, respectively.
(PPTX)

**S5 Table. List of residues interacting with the docked cis-1,4-polyisoprene, extracted from PLIP analysis of 10 docking poses obtained using ChemScore.** Docking solutions were ranked based on the fitness score from highest to lowest. Residues that interact with all / most poses are highlighted in grey and light grey, respectively.
(PPTX)

**S6 Table. Frequency of contacts between the substrate and LcpK30 calculated during 100 ns MD simulations of 10 docking poses obtained from ChemPLP fitness function.** Docking solutions were initially ranked based on the fitness score from highest to lowest. A cut-off of 5 Å was used and only contacts with total fraction more than 4.5% are listed. Ext = extended conformation; Fold = folded conformation.
(PPTX)

**S7 Table. Frequency of contacts between the substrate and LcpK30 calculated during 100 ns MD simulations of 10 docking poses obtained from ChemScore fitness function.** Docking solutions were initially ranked based on the fitness score from highest to lowest. A cut-off of 5 Å was used and only contacts with total fraction more than 4.5% are listed. Ext = extended conformation; Fold = folded conformation.
(PPTX)

## Acknowledgments

We would like to thank Prof. Derek Irvine for his contribution and useful discussion on project direction. We also gratefully acknowledge the support and access to the University of Nottingham High Performance Computing Facility.

## Author Contributions

**Conceptualization:** Aziana Abu Hassan, Anca Pordea.

**Formal analysis:** Aziana Abu Hassan, Marko Hanževački.

**Funding acquisition:** Aziana Abu Hassan, Anca Pordea.

**Investigation:** Aziana Abu Hassan, Marko Hanževački.

**Methodology:** Aziana Abu Hassan, Marko Hanževački.

**Software:** Marko Hanževački.

**Supervision:** Anca Pordea.

**Validation:** Aziana Abu Hassan.

**Visualization:** Marko Hanževački.

**Writing – original draft:** Aziana Abu Hassan, Marko Hanževački, Anca Pordea.

**Writing – review & editing:** Marko Hanževački, Anca Pordea.

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
