## [Decision Letter · Decision Letter 0]

14 Sep 2023

PONE-D-23-23541Computational investigation of cis-1,4-polyisoprene binding to the latex clearing protein LcpK30PLOS ONE

Dear Dr. Pordea,

Thank you for submitting your manuscript to PLOS ONE. After careful consideration, we feel that it has merit but does not fully meet PLOS ONE’s publication criteria as it currently stands. Therefore, we invite you to submit a revised version of the manuscript that addresses the points raised during the review process.

We look forward to receiving your revised manuscript.

Kind regards,

Sushil Mishra, Ph.D

Academic Editor

PLOS ONE

Journal Requirements:

Reviewers' comments:

Reviewer's Responses to Questions

**Comments to the Author**

1. Is the manuscript technically sound, and do the data support the conclusions?

Reviewer #1: Partly

Reviewer #2: No

2. Has the statistical analysis been performed appropriately and rigorously? 

Reviewer #1: N/A

Reviewer #2: I Don't Know

3. Have the authors made all data underlying the findings in their manuscript fully available?

Reviewer #1: Yes

Reviewer #2: Yes

4. Is the manuscript presented in an intelligible fashion and written in standard English?

Reviewer #1: Yes

Reviewer #2: Yes

5. Review Comments to the Author

Reviewer #1: The studied enzymes, latex clearing proteins (Lcps) are interesting candidates for structural studies as they can cleave polymers despite having active catalytic site buried inside the protein structure and furthermore, they can be used to degrade of rubbers and potentially even synthetic polymers. Application of various computational tools, such as CAVER-Pymol plugin 3.0.3, fpocket and Molecular Dynamic (MD) simulations allowed the authors to analyse the structures of Lcps and substrate binding.

Comment 1

The authors compared the open and closed crystal structures of LcpK30. The open crystal structure of LcpK30 contains, apart from heme, an imidazole molecule in its active site. The open crystal structure in single state is thoroughly analysed by the authors, also considering the probability of acquisition of a polymer in the cavities. It is not confirmed, however, if the analysed protein would have the same open conformation during incorporation of a large oligomeric substrate, as it has while incorporating imidazole, which is a smaller aromatic molecule. In PET-degrading enzymes that have active site exposed on the surface, it is the binding site’s flexibility that most probably allows the enzyme to incorporate such polymer substrate for depolymerisation. The other example can be found in epoxide hydroxylases, the native long hydrophobic substrate was thought to enter through one tunnel and with proper positioning was reaching exit of the second tunnel. Both tunnels made a L-shape tunnel network. However, during MD study, it was shown, that two domains of the epoxide hydrolase can undergo small conformational changes, and instead of two isolated tunnel, groove can be observed (https://doi.org/10.1016/j.csbj.2021.10.042 ). It is thus possible that LcpK30 undergoes some strong conformational changes to allow easier accommodation of oligomeric substrate to the protein core where the active site is located. Such big conformational changes were not possible to be observed by the authors in the performed studies. The results of the docking studies using GOLD software of flexible model substrate to rigid protein suggest that the protein truly needs to undergo some conformational changes prior to incorporating the substrate, as the obtained poses had very low or even negative fitness score.

Comment 2

The authors identified putative tunnels within the crystal structure of LcpK30 only in the OPEN state, using the CAVER-PyMOL plugin. CAVER-PyMOL plugin allows to analyse tunnels in the macromolecule’s core using the geometric approach in single state. The results are strongly influenced by the chosen radius probe. The authors should probably also analyse LcpK30 in the CLOSED state for comparison using CAVER-PyMOL plugin. Moreover, since authors run MD simulations, it could be recommended to use either CAVER implementation for MD simulation analysis or AQUA-DUCT which can more precisely show the potential accessing paths. Authors need to take into consideration, that CAVER and specifically CAVER plugin is using radial probe for tunnels detection and analysis, therefore all asymmetric tunnels cannot be described properly. Instead some of the tunnels has to be merged together. Such problems can be avoided e.g. using water molecules as a small probes – this is especially important for bulky substrates, or substrates which are not globular.

Authors can see, what type of analysis is accessible in previously mentioned article https://doi.org/10.1016/j.csbj.2021.10.042 and comparison of geometry based approach (CAVER) with molecules tracking approach (AQUA-DUCT) can be found https://doi.org/10.1021/acs.jcim.2c00985 and https://doi.org/10.1371/journal.pcbi.1010119 In general, the analysis of tunnels in crystal structures only can be misleading. Since authors have MD simulations in hands, the analysis of accessibility to the active site can be done quickly with small effort only.

Comment 3

Plot in fig.3 a should not be a line plot, but a point plot.

Comment 4

In lines 504-507 and 567-570 the authors mention experimental results of distribution of oligomers produced by LcpK30. It is not clear however, what results they mean exactly (citation?) and what are the oligomers produced by LcpK30 and which part of the modelled substrate they constitute.

Comment 5

In lines 589-592 of the methods section, the authors mention that they remove water molecules from the crystal structure of LcpK30. Then in lines 677-680 they describe the solvation of protein with OPC water using the truncated octahedron box. Prior to solvation, the authors should add the crystal water as to ensure the correct water networks inside the protein core. In the case of holo system, only crystallographic waters producing clashes with the docked substrate, should have been removed. Such procedure is much safer, and the equilibration time is shorter. In case when whole buried water is removed from the interior, some cavities might initially be closed, and longer time is required to expand them to size accommodating water again. Authors can also used 3D-RISM and Placevent for system preparation. More details can be found in papers dedicated to mentioned methods or in review describing methods using water molecules in computational analysis (https://doi.org/10.1016/j.csbj.2020.02.001 ).

Overall, the study gives interesting insight into LcpK30 structure, tunnel and pocket network and their dynamics, and provides hypotheses on how Lcps can incorporate and cleave oligomeric substrates. The authors identified probable tunnels used for transport of the oligomeric substrates to the active site. However, there are doubts if analysis was done deep enough to provide true description of the substrate accessibility.

Article provide important findings, as the tunnel-lining residues are suitable targets for mutagenesis aimed at improving enzymatic degradation rates. The information on tunnels structure and dynamics mostly comes from analysis of open structure of LcpK30. However, the MD simulations should provide insight into much more opened structure. Authors has mentioned helix E as important one – on the second side of the hem, we can see C-terminal loop, which is stabilised by short helix (71-81). It is quite probable, that this sandwich can open more significantly. The second tunnel will remain separated by the loop. Authors has identified I396 as a one of the interacting residues – it is the one which separates two distinctive tunnels from CAVER analysis.

The authors could compare the closed structure as well and tunnels dynamics in the structures with bound substrate to more deeply understand what conformational changes the LcpK30 enzyme undergoes and how incorporation of a big oligomeric substrate influences tunnel dynamics. This was partly done by RMSD and RMSF analysis, but tunnel radius investigation in a enzyme-substrate complex could give insight into tunnel bottlenecks and residues that might interfere in the substrate/product transport.

I would like to underlay that since authors have run MD simulations the mentioned improvements are in their hands. However, the conclusions can differ substantially.

Reviewer #2: The manuscript by Hassan and co-workers brings an interesting approach to the investigation of the polyisoprene binding to latex-clearing proteins and indeed might represent a good contribution for a better understanding of the mechanisms behind the action of this type of protein. The submitted version of the manuscript, however, seems to show some flaws that don't permit ensuring publication. Some suggestions of improvements are listed below:

1) The english language is good but small typos and grammar issues were detected. I recommend a deep revision before resubmission;

2) I don't see need of an author summary. Unless this is a request from the journal I suggest deleting it;

3) I suggest no repeating colors in Figs. 2a and 2b. This can confuse the readers. Use different colors for tunnels and pockets;

4) The docking protocol is poor. There is no re-docking to validate the docking protocol neither an appropriate description about the number of runs, criteria to select the best poses, algorithm used, etc.... Also there is no experimental support for the binding of the ligand. Authors should use some experimental evidence that justify the poses chosen. If it's not possible, this should be clarified in the text and the speculative character of the work evidenced;

5) How many poses were selected for the MD simulations? this is not clear in the manuscript. Why to select folded and extended poses? There is no hint in the literature pointing to one or another? This should be discussed in the text;

6) Most docking programs state favorable docking scores as negative energy values. If GOLD is different this should be clarified in the text;

7) Many important results are shown as supplementary material. This makes it hard to follow the discussion. I recommend revising it and bringind back to the text the most relevant figures and tables.

8) Most of software and methods used are not properly cited in the manuscript;

9) Author mention binding energy calculations during the MD simulations but don't present any result.

6. PLOS authors have the option to publish the peer review history of their article (what does this mean?). If published, this will include your full peer review and any attached files.

Reviewer #1: No

Reviewer #2: No

---

## [Author Response · Author response to Decision Letter 0]

1 Feb 2024

Thank you for giving us the opportunity to respond to reviewers comments and to revise our manuscript. We carefully considered and addressed the editor’s and reviewers’ comments and include our response in the attached document "Response to reviewers". We thank the reviewers for their very insightful comments, which allowed us to improve the manuscript.

Journal Requirements

Response: we used the links above to format our documents and included all supplementary figures and tables as separate files, named and cited appropriately in the text.

Response: funding text was removed.

Response: the relevant accession number for all the data was provided as a link included in the Data Availability Statement at the end of the document. https://doi.org/10.6084/m9.figshare.22941149.

Reviewers' comments

We thank the reviewers for their very insightful comments, which allowed us to improve the manuscript. We addressed all comments below. We included additional data and explanations and we moved some of the data from the supplementary information into the main manuscript, as suggested by the reviewers. 

Reviewer #1

The studied enzymes, latex clearing proteins (Lcps) are interesting candidates for structural studies as they can cleave polymers despite having active catalytic site buried inside the protein structure and furthermore, they can be used to degrade of rubbers and potentially even synthetic polymers. Application of various computational tools, such as CAVER-Pymol plugin 3.0.3, fpocket and Molecular Dynamic (MD) simulations allowed the authors to analyse the structures of Lcps and substrate binding.

Comment 1.1 

The authors compared the open and closed crystal structures of LcpK30. The open crystal structure of LcpK30 contains, apart from heme, an imidazole molecule in its active site. The open crystal structure in single state is thoroughly analysed by the authors, also considering the probability of acquisition of a polymer in the cavities.

It is not confirmed, however, if the analysed protein would have the same open conformation during incorporation of a large oligomeric substrate, as it has while incorporating imidazole, which is a smaller aromatic molecule.

In PET-degrading enzymes that have active site exposed on the surface, it is the binding site’s flexibility that most probably allows the enzyme to incorporate such polymer substrate for depolymerisation. The other example can be found in epoxide hydroxylases, the native long hydrophobic substrate was thought to enter through one tunnel and with proper positioning was reaching exit of the second tunnel. Both tunnels made a L-shape tunnel network. However, during MD study, it was shown, that two domains of the epoxide hydrolase can undergo small conformational changes, and instead of two isolated tunnel, groove can be observed (https://doi.org/10.1016/j.csbj.2021.10.042 ). It is thus possible that LcpK30 undergoes some strong conformational changes to allow easier accommodation of oligomeric substrate to the protein core where the active site is located.

Such big conformational changes were not possible to be observed by the authors in the performed studies. The results of the docking studies using GOLD software of flexible model substrate to rigid protein suggest that the protein truly needs to undergo some conformational changes prior to incorporating the substrate, as the obtained poses had very low or even negative fitness score.

Response:

We thank the reviewer for this question because it allowed us to further discuss the importance of the conformational flexibility in LcpK30. To address this question, we carried out extensive 3 x 500 ns MD simulations starting from the closed-like conformation of the enzyme, prepared as described below. In addition to that, to explore conformational changes of the enzyme upon enzyme-substrate complex formation, we performed principal component analysis (PCA) taking into account backbone cartesian coordinates, using MD snapshots from both the open and the closed-like states of the empty enzyme (without substrate bound), as well as of the enzyme in complex with the polyisoprene substrate. New text and data was included in the manuscript, as described below.

Pages 8-10: we introduced changes in title, text and Fig 2 title to clarify differences between static and dynamic models.

Lines 189-196

“We assumed that polyisoprene binding occurs in the catalytically active state of LcpK30 and therefore we prepared an LcpK30 model starting from the open state structure, in which the co-crystalized imidazole molecule was replaced with O2 bound to the heme cofactor and where the Lys167 chain (not heme bound) was fully protonated, which we were able to simulate during relatively long simulation timescales. For comparison, we also modelled the closed-like state system, prepared from the closed state structure conformation, but where the coordination of Lys167 to heme was removed and replaced with O2 coordination, whilst Lys167 was fully protonated.”

We note that the crystal structure of the closed conformation (PDB 5O1M) most likely resembles the resting state of the enzyme, with neutral Lys167 (instead of O2) coordinating heme. We were not interested in simulating this state because the coordination of Lys167 to Fe would prevent binding of the catalytic O2 and its subsequent interaction with the substrate.

Lines 198-202

“Relatively low Calpha backbone RMSD were obtained over the course of 500 ns MD simulations, for both open and closed-like conformations (see S3a Fig for the open conformation). This suggested that the enzyme with no substrate bound did not undergo large structural changes compared to the reference X-ray structures of the open and closed states.” 

Lines 212-244

“Principal component analysis (PCA) taking into account backbone cartesian coordinates was used to compare open and closed states of the enzyme with no substrate. A clear separation of the two states along the PC1 was observed, suggesting that major conformational changes must occur that would allow the transition between closed (positive PC1 values) and open (negative PC1 values) conformations (Fig 3a). Furthermore, the normal mode displacements (Fig 3b) along the PC1 were characterised as conformational changes that predominantly involve helix E residues 160-176, which also contains the central Lys167. Namely, in the open conformation Lys167 is exposed towards the protein surface interacting with the propionates of the heme cofactor, in contrast to the closed-like conformation where it is rotated into the protein interior interacting with O2 and Glu148. On the other hand, displacements along the PC2 are associated with the flexibility of the C-terminus loop 392-397 ranging from fully closed (negative PC2 values) to a fully open (positive PC2 values) conformation (Fig 3b).

Despite the closed-like state having identical amino acid protonation states and the same species present in the active site as the open state (fully protonated Lys167 and O2 bound to heme), a transition between the two states has not been observed, and the reason for that might be a large barrier separating these two states that simply could not be sampled using conventional MD simulations at the available timescales. This implies that the opening of the enzyme is a complex and controlled process that happens on longer timescales and involves large conformational changes of the helix E and Lys167 and might even be coupled with the approach and the interaction with the polyisoprene substrate.”

We also introduced a principal component analysis to compare the conformations of LcpK30 with and without bound substrate. For this, we combined the MD snapshots for the empty enzyme (no substrate, from analysis above) with MD snapshots of the enzyme in complex with the cis-1,4-polyisoprene chain (originating from all simulations of both ChemPLP and ChemScore docking solutions) into a unique PCA plot, now included in S13 Fig. Based on the PCA plots, we concluded the following:

Lines 437-451

“The PCA analysis combining MD snapshots of the enzyme with and without substrate showed that compared to the empty enzyme, the presence of substrate slightly changed the conformation of the enzyme, irrespective of the extended or folded substrate binding mode (S13 Fig). The bound and unbound states were not separated along only one principal component, instead displacements along both PC1 and PC2 were observed. Normal mode displacements along the PC1 included the loop 155-160, helix E 160-176 and C-terminal helix Z2 297-314, whilst the displacement along the PC2 mostly localised at the C-terminal loop residues 392-397. As observed from the RMSD analysis, in some cases where the substrate binds deeper in the active site the conformation of the enzyme bound to substrate was significantly different from the empty enzyme (for example in the case of extended substrate conformations 10 from ChemPLP and 3 from ChemScore). Furthermore, significant structural changes of the enzyme, especially C-terminus loop 392-397, were observed in case of simulations with the folded substrate conformations 1 from both ChemPLP and ChemScore.”

 From this data, we conclude that conformational changes occur for the substrate-bound enzyme, compared to either open or closed state without substrate. Simulation of the closed-like state with substrate bound was not possible, because the substrate could not be accommodated within the structure. However, we hypothesise that the binding of the substrate is more likely to occur to the open state than to the closed state, because more space is available within the enzyme (see also answer below, related to tunnels in closed structure). 

The reviewer also suggested that there might be a groove forming which would allow easier approach and binding of larger substrates into LcpK30, similar to the one reported for epoxide hydrolase, however we did not observe the formation of such groove from our simulations. We believe that in the case of LcpK30, the tunnel formation is most likely controlled by the conformation of central Lys167 from the helix E and its interaction with the heme propionate groups and by the conformation of the flexible C-terminus loop.

Comment 1.2 

The authors identified putative tunnels within the crystal structure of LcpK30 only in the OPEN state, using the CAVER-PyMOL plugin. CAVER-PyMOL plugin allows to analyse tunnels in the macromolecule’s core using the geometric approach in single state. The results are strongly influenced by the chosen radius probe. The authors should probably also analyse LcpK30 in the CLOSED state for comparison using CAVER-PyMOL plugin. Moreover, since authors run MD simulations, it could be recommended to use either CAVER implementation for MD simulation analysis or AQUA-DUCT which can more precisely show the potential accessing paths. Authors need to take into consideration, that CAVER and specifically CAVER plugin is using radial probe for tunnels detection and analysis, therefore all asymmetric tunnels cannot be described properly. Instead some of the tunnels has to be merged together. Such problems can be avoided e.g. using water molecules as a small probes – this is especially important for bulky substrates, or substrates which are not globular. Authors can see, what type of analysis is accessible in previously mentioned article https://doi.org/10.1016/j.csbj.2021.10.042 and comparison of geometry based approach (CAVER) with molecules tracking approach (AQUA-DUCT) can be found https://doi.org/10.1021/acs.jcim.2c00985 and https://doi.org/10.1371/journal.pcbi.1010119 In general, the analysis of tunnels in crystal structures only can be misleading. Since authors have MD simulations in hands, the analysis of accessibility to the active site can be done quickly with small effort only.

Response:

No static tunnels were identified using CAVER-Pymol plugin 3.0.3 within the closed state structure (5O1M), using the same calculation parameters as with the open state structure. However, by decreasing the minimum probe radius of tunnel calculations from 0.9 to 0.6 Å, three tunnels were detected, with much lower throughput (0.2 Å) and bottleneck radius (0.6 Å). 

We included a clarification at lines 132-133: “We used both the open (5O1L) and the closed state conformations (5O1M), however only the former led to tunnel identification using the default parameters (see Methods).”

We thank the reviewer for suggesting alternative programs for detecting and analysing tunnels from MD simulations. We are aware of the limitations of CAVER software when it comes to characterising wide and asymmetric tunnels, which arise from the fact that the individual tunnels are detected by constructing the Voronoi diagrams using simple spherical probes. However, we strongly believe that accurate tunnel predictions could be obtained by employing the dynamical approach implemented in CAVER Analyst 2 using MD snapshots instead of static crystal structures. In the previously submitted S5a Fig and S2 Table, we have already presented the dynamical tunnel calculations with CAVER Analyst 2 using our MD simulations of the enzyme in open conformation. 

We have now moved the analysis of dynamical tunnel clusters for the open state enzyme from the supplementary information into the main manuscript (Table 1 and Fig 4a). In addition to this, we also analysed the features of dynamical tunnel clusters of the closed-like conformation and compared those to the tunnels obtained during MD simulations of open state (Table 1 and S5 Fig). New text and data was included in the manuscript, as described below.

Lines 245-290:

“Given the enzyme flexibility required to accommodate the substrate, we sought to improve the static tunnel predictions by employing the dynamical approach implemented in CAVER Analyst 2, using MD snapshots instead of static crystal structures. The first 5 identified tunnel clusters ranked based on priority (calculated by averaging tunnel throughputs over all snapshots) were analyzed, for MD simulations with both the open and closed-like conformations and the details about the features of the dynamic tunnel clusters are shown in Table 1. Open state tunnel clusters had overall larger average and maximum bottleneck radius, shorter length, smaller curvature and higher throughput than closed state ones, which ultimately makes them more favorable for substrate binding. Furthermore, the priority of the tunnel clusters calculated based on the appearance frequency during the MD simulations was much higher in the open (14-49%) compared to the closed (4-13%) state. Interestingly, when comparing the spatial distribution of timeless tunnel cluster pathways, we could clearly see how the central Lys167 plays an important role in separating tunnels into two wider superclusters in the open state (Fig 4a, clusters 1 and 3 forming the green supercluster and clusters 2 and 5 forming the blue super-cluster). This is in contrast with the rotated Lys167 conformation forcing the appearance of one narrow supercluster 

---

## [Editor Report · Decision Letter 1]

3 Apr 2024

Computational investigation of cis-1,4-polyisoprene binding to the latex clearing protein LcpK30

PONE-D-23-23541R1

Dear Dr. Pordea,

We’re pleased to inform you that your manuscript has been judged scientifically suitable for publication and will be formally accepted for publication once it meets all outstanding technical requirements.

Kind regards,

Sushil Mishra, Ph.D

Academic Editor

PLOS ONE

Additional Editor Comments (optional):

None